# "Grumpy" or "furious"? arousal of emotion labels influences judgments of facial expressions

**Megan S. Barker** [1,2]*, **Emma M. Bidstrup**[1], **Gail A. Robinson**[1,3], **Nicole L. Nelson**[1]

**1** School of Psychology, The University of Queensland, St Lucia, Brisbane, QLD, Australia, **2** Department of Neurology, Taub Institute for Research on Alzheimer's Disease and the Aging Brain, Gertrude H. Sergievsky Center, Columbia University Medical Center, New York, NY, United States of America, **3** Queensland Brain Institute, The University of Queensland, St Lucia, Brisbane, QLD, Australia

* msb2228@cumc.columbia.edu

**Data Availability Statement:** The data that support the findings of this study have been made publicly available via Open Science Framework. Data can be accessed at https://osf.io/xc9a5

## Abstract

Whether language information influences recognition of emotion from facial expressions remains the subject of debate. The current studies investigate how variations in emotion labels that are paired with expressions influences participants' judgments of the emotion displayed. Static (Study 1) and dynamic (Study 2) facial expressions depicting eight emotion categories were paired with emotion labels that systematically varied in arousal (low and high). Participants rated the arousal, valence, and dominance of expressions paired with labels. Isolated faces and isolated labels were also rated. As predicted, the label presented influenced participants' judgments of the expressions. Across both studies, higher arousal labels were associated with: 1) higher ratings of arousal for sad, angry, and scared expressions, and 2) higher ratings of dominance for angry, proud, and disgust expressions. These results indicate that emotion labels influence judgments of facial expressions.

## Introduction

The ability to accurately judge the emotional state of others is a vital skill for everyday life. Over the past several decades, researchers have focused on the role of facial expressions in communicating emotion, and, indeed, facial expressions directly inform our interpretations of others' behaviour. However, recent research has begun to examine the various forms of information that we use to make emotion judgments.

### Perspectives on emotion

There exist several perspectives regarding how faces communicate emotion, and how observers subsequently perceive emotion from faces [1]. The traditional *basic emotions approach* suggests that emotions are automatically and universally recognised via discrete facial expression categories, and this recognition is unaffected by contextual information (i.e., anger, fear, surprise, sadness, disgust, contempt, happiness) [2–4]. However, people do not consistently produce the hypothesised expressions in daily life and recognition across cultures is poor [5–11].

**Funding:** At the time of writing, author M.B. was supported by an Australian Government Research Training Program Scholarship.

**Competing interests:** The authors have declared that no competing interests exist.

As an alternative view, the *constructionist approach* argues that emotions are not perceived via the interpretation of automatic and innate facial muscle movements (i.e. facial expressions), but rather that emotions are constructed in the minds of the perceivers [7,12–14]. This view asserts that language plays a constitutive role in emotion perception [7,15]. Constructionists argue that facial expressions are ambiguous, varied, and not prototypically representative of their underlying emotions, and so emotion words provide the context that draws meaning between them [7,12,15]. For example, someone might witness an individual scowl when he receives a speeding fine, purse his lips at undue criticism, and clench his jaw at his child misbehaving. These instances share almost no perceptual similarities and would therefore not form a coherent category without the word "anger" to connect them. An array of emotion words is available in the minds of healthy adults at all times, and these words form a "context" that allows facial expressions (along with situational clues) to be transformed into perceptions of emotion. Thus, Barrett et al. [16] suggest that language information (i.e., emotion words) encourages perceivers to engage in "top-down" processing, incorporating information in order to settle on a more informed judgment of emotion.

## Language influences emotion judgments

The relevance of language to emotion judgments has been repeatedly demonstrated. For example, providing participants with a list of emotion words to match to expressions (i.e., a forced-choice as opposed to free-labelling task) increases accuracy by up to 26% [17–20], whereas including irrelevant or incongruent emotion words leads to decreased accuracy of judgments [21–23]. In addition, bilingual participants are more likely to match emotion labels with their hypothesised expression when tested in English as opposed to their native language [see 8 for a review,24,25]. When presented with a novel emotion label, children and adults match the novel label to a novel expression [26–28]. These studies provide converging evidence that emotion words can play a powerful role in our judgments of emotion.

The *linguistic relativity hypothesis* suggests that language shapes the thoughts of its users [29–31]. When children learn language they also acquire important knowledge about how we conceptualise and categorise the world [32]; this includes emotion knowledge [33,34]. Conceptual knowledge and language are intrinsically linked [35]; therefore, when language for a specific concept is absent or inaccessible to the individual, its conceptual meaning is lost [36]. For example, semantic satiation of emotion words (i.e., presenting the emotion word 30+ times to make conceptual knowledge of the emotion temporarily inaccessible) leads to greater difficulty in identifying emotion from facial expressions [12,37]. Furthermore, individuals with semantic dementia (a neurodegenerative disorder in which concept knowledge is degraded) have difficulty sorting facial expressions into discrete emotion categories [38]. These data suggest that when conceptual knowledge of emotion categories is disrupted, emotion judgments are poorer, highlighting the importance of language in emotion recognition processes.

The effect of language as a contextual influence on emotion perception has also garnered a solid base of research. The paradigm commonly focuses on matching facial stimuli to relevant emotion words or categories [39–41]. Emotion words provide information that influences precision and accuracy of emotion judgments, causing a shift in the way that faces are perceived [42]. Pairing emotional faces with labels increases sensitivity and speed in recognising emotions, even in participants with alexithymia (who are impaired in labelling their own emotions) [43]. Along similar lines, verbal explanations of emotion can influence memory of facial expressions [44–46]. Recently, Fugate et al. [47] reported that participants were faster to detect a change in a transitioning face (e.g., moving from anger to disgust) when an emotion word congruent with the first emotion was presented, and slower when an incongruent word was

presented [47]. Similarly, the same research group showed that presenting emotion words following a target face shifted perception of the face (that is subsequently selecting the target face from an array) to align with the word, especially when exposure to the target face was limited [41].

## Movement as contextual information

The influence of movement in judging others' expressions can also provide important context, and better approximates the moving expressions observed in daily life [48]. Previous research has shown that dynamic expressions provide more information than static expressions and influence how features of expressions are visually processed [49–53].

## The current studies

The current studies aim to investigate whether emotion word labels are integrated into judgments of emotion in facial expressions. To ensure we examined broad patterns in judgements of facial expressions, we incorporated a variety of facial expressions. We also measured participants' emotional judgments via dimensions of affect (e.g., valence, arousal, dominance) [54–57], rather than discrete categories. The influence of emotion words on perceived dimensional affect has been less examined than discrete emotion categories, and "categorical thinking", compared to "shades of grey", may shift emotion judgments (e.g. choosing "fear" or "calm" vs. a sliding scale from fear to calm) [58]. It is possible that the reason language context influences perception of discrete emotion categories is because it shifts participants' perceptions of the underlying affective dimensions (i.e., arousal, valence, or dominance). Here we investigate whether linking facial expressions with emotion labels of varying arousal levels will influence participants' affective judgments (arousal, valence, dominance) of facial expressions.

Unlike previous research, where the emotion words used represented discrete emotion categories [43], the labels used in the current studies are synonyms within emotion categories that vary in their level of arousal (e.g., low and high arousal synonyms of "anger" are "grumpy" and "furious", respectively; synonyms were selected based on a pilot study described later). We opted to manipulate arousal in emotion labels (rather than valence) because the large database of participant word ratings we planned to use [59] showed that arousal ratings were more variable across words than valence ratings.

We also included measures of perceived valence and dominance in the current studies because emotion labels that vary in arousal are likely to vary in valence and dominance as well [59,60]. For example, the emotion word "furious" is higher in arousal than "grumpy" and may also convey more negative valence (unpleasantness) and increased dominance (powerfulness). Warriner et al. [60] obtained arousal, valence and dominance ratings of 13,915 English words, and reported a U-shaped relationship between arousal and valence. That is, they found a positive correlation between valence and arousal for positive words (i.e., more positive valence = higher in arousal) and a negative correlation between valence and arousal for negative words (i.e., more negative valence = higher in arousal; Warriner et al., 2013). This finding is well established through the literature [e.g., 59,61,62]. Likewise, Warriner et al. [60] reported a U-shaped relationship between arousal and dominance (i.e., a positive correlation between arousal and dominance for high dominance words and a negative correlation for low dominance words). Therefore, including measures of perceived valence and dominance will allow us to more thoroughly explore how pervasive the effect of emotion label arousal is on affective perception of faces.

In Study 1, traditional static stimuli were used. Study 2 extended the findings of Study 1 to dynamic (moving) facial expression stimuli. We predicted that label arousal level would

influence arousal ratings of the associated expressions, such that faces paired with high arousal labels will be rated higher in arousal than faces paired with low arousal labels. Moreover, we hypothesised that label arousal level would also influence the valence and dominance ratings of faces consistent with the valence and dominance of the label. For example, we expected that if a high arousal label was low in valence, the valence judgement of the paired facial expression would be lowered.

## Study 1: Static stimuli

### Method

**Participants.** A sample of 172 psychology students at the University of Queensland was recruited through the School of Psychology research participation scheme, and participants received course credit for their participation. The sample comprised 90 females and 82 males, aged 17 to 53 years ($M$ = 20.5 years). English was the native language of 129 participants and all participants were proficient in English to a tertiary education level. In line with similar research in this area, our goal was to collect at least 50 participants per condition [41,47]. This study was approved by the University of Queensland School of Psychology Research Ethics Committee (approval number 16-4-56-AH). All participants provided informed, written consent to participate.

An additional 18 participants were recruited via word-of-mouth to participate in a pilot study. This sample included 13 females and 5 males aged between 18 and 65 years. Seventeen pilot participants indicated that English was their primary language.

### Materials

**Facial expressions.** Thirty-two facial expressions were selected from the Amsterdam Dynamic Facial Expressions Set (ADFES) [63]. The ADFES expressions are standardised and coded using the Facial Action Coding System [64]. In a validation study, all pressions in the set were identified by participants as the intended emotion in 74% of trials [63].

We created static stimuli by capturing the apex of each expression as a static image. We selected four expressions (two male posers and two female posers) for each of the eight expressions included: happy, sad, angry, scared, disgusted, embarrassed, proud and surprised. The four expressions for each emotion were drawn from posers within the ADFES set, with some posers repeating across emotion categories, but no poser displayed expressions for all emotions included. The posers varied across emotion categories. The images were presented at a resolution of 720 x 576.

**Emotion word labels.** Twenty-four emotion labels were used in this study (see Table 1), selected according to piloted arousal ratings. Initially, 67 emotion words representing synonyms for eight emotion categories (happy, sad, angry, scared, disgusted, embarrassed, proud and surprised), with approximately eight labels for each emotion category, were selected from both the Affective Norms of English Words set [59] and from a list of affective words generated by participants in previous research [65,66]. Eighteen pilot participants then rated the arousal level of each label on a Likert-type scale (1.00 = *very sleepy* and 7.00 = *very awake)* (all pilot data are available in the Supplemental Materials, S1 Appendix). Three labels were selected for each of the eight emotions based on the pilot results: those with the lowest ($M$ = 3.60), most moderate ($M$ = 4.50) and highest ($M$ = 5.60) arousal ratings of each emotion category.

### Procedure

Participants were randomly assigned to view one of three conditions: Faces+Labels, Labels Alone, or Faces Alone. Those assigned to the Faces+Labels condition ($N$ = 53) were presented

**Table 1. Means and standard deviations of arousal ratings for low and high arousal labels alone for each emotion category.**

| Emotion | Label | Arousal Level | *M* | *SD* |
|---|---|---|---|---|
| Happy | Contented | Low | 3.98 | 1.14 |
| | Elated | High | 5.33 | 1.43 |
| Sad | Down | Low | 2.91 | 0.82 |
| | Distraught | High | 4.36 | 1.61 |
| Angry | Grumpy | Low | 3.57 | 1.37 |
| | Furious | High | 6.13 | 0.98 |
| Scared | Worried | Low | 4.59 | 1.28 |
| | Terrified | High | 5.70 | 1.33 |
| Disgusted | Nauseated | Low | 3.73 | 1.19 |
| | Repulsed | High | 4.89 | 1.15 |
| Embarrassed | Ashamed | Low | 4.23 | 1.01 |
| | Mortified | High | 5.15 | 1.23 |
| Proud | Satisfied | Low | 4.40 | 1.31 |
| | Victorious | High | 5.86 | 1.03 |
| Surprised | Awed | Low | 4.95 | 1.24 |
| | Astounded | High | 5.53 | 1.24 |

emotion word labels within the phrase, "This person is feeling ___________", which was subsequently removed from the screen and followed by a facial expression from the same overarching emotion category (e.g., the label "elated", followed by a "happy" face). Participants were explicitly instructed to rate the face only. Participants rated each of the four faces paired with each of the three labels within the same emotion category (e.g., each happy face paired with "contented", "pleased", and "elated"). This resulted in 96 trials overall (12 trials for each emotion), presented to participants in a randomised order. The experiment was run using Qualtrics, and responses were self-paced with participants allowed to take as long as they liked to answer the questions. Additionally, 48 participants were assigned to rate the 24 labels (8 emotions x 3 arousal levels) in isolation (i.e., without faces), and 71 participants were assigned to rate the 32 faces (8 emotions x 4 posers) in isolation (i.e., without labels).

Participants in all conditions rated the stimuli across three affective dimensions using a series of 7-point Likert scales. The three scales were presented on a single page and in the following order: valence (1.00 = *very negative*, to 7.00 = *very positive*), arousal (1.00 = *very sleepy*, to 7.00 = *very awake*), and dominance (1.00 = *very weak*, to 7.00 = *very powerful*) (see Supplemental Materials, S2 Appendix for an example of the test trial format. Participants rated four posers' expressions for each emotion label, thus we averaged participants' ratings across faces for each emotion label.

## Results

### Labels manipulation check

Results from the Labels Alone condition ($N = 48$) revealed that the arousal ratings of labels were generally similar to those elicited in the pilot study in that they reflected the expected arousal order for high and low arousal labels (i.e., high arousal labels were rated significantly higher in arousal than low arousal labels for all emotion categories, all Bonferroni-corrected $p < .05$). However, medium arousal labels were inconsistently rated as moderate in arousal, relative to low and high arousal labels. Therefore, we excluded medium arousal labels from all subsequent statistical analyses, as their inconsistent arousal ratings violated the assumptions of the

model (for all label ratings and analyses with all labels see Supplemental Materials, Appendices C, D, E).

Labels were chosen based on arousal ratings, as this was the primary manipulation in the study. However, as previous research has shown that arousal, valence and dominance of words vary together, valence and dominance ratings were also collected for the low and high arousal labels (see Supplemental Materials, S3 Appendix). Paired-samples *t*-tests with Bonferroni-corrected *p* values revealed that compared to low arousal labels, high arousal "happiness" and "pride" labels were rated more positively (i.e., higher in valence) and high arousal "sadness", "fear", "disgust" and "surprise" labels were rated more negatively (i.e., lower in valence), all *p* < .05. High vs. low arousal labels for "happiness" "anger" "embarrassment" and "pride" were not significantly different in terms of valence, all *p*s > .05. Similarly, compared to low arousal labels, high arousal "anger", "disgust", "embarrassment" "pride" and "surprise" labels were rated as higher in dominance, all *p* < .05. High vs. low arousal labels for "happiness" "sadness" and "fear" were not significantly different in terms of dominance, all *p*s > .05.

## Main analyses

To investigate the influence of labels on perceived affect of emotional faces, we conducted an 8 (emotion category) x 2 (label arousal level) repeated-measures ANOVA on the Faces+Labels data for each dimension measured (arousal, valence, dominance). We applied Greenhouse-Geisser corrections for analyses in which assumptions of sphericity were violated. Significant interactions were followed up with paired-samples *t*-tests, and Bonferroni-corrected *p* values are reported for these analyses. In the following sections we specify only the significant effects and contrasts. Although not the primary focus of our study, we opted to further investigate the effect of labels on perceived arousal of faces by comparing ratings of faces paired with labels (Faces+Labels condition) with ratings of Faces Alone. These analyses are available in the Supplemental Material, S6 Appendix. All main analyses were re-run with non-native English speakers excluded and the results were unaffected, so all participants are included here.

## Arousal

As predicted, arousal ratings for faces paired with labels differed between emotion categories and label arousal level. There was a main effect for emotion category, $F(4.49, 233.25) = 56.59$, $p < .001$, $\eta p2 = .521$, and a main effect for label arousal level, $F(1, 52) = 62.14$, $p < .001$, $\eta p2 = .544$. There was also a significant two-way interaction, which indicated that the effect of label arousal level on perceived arousal of faces varied by emotion category, $F(4.67, 242.80) = 4.97$, $p < .001$, $\eta p2 = .087$. Post hoc analyses revealed that ratings for sad, angry, scared and proud faces differed significantly depending on the paired label (see Fig 1). Sad faces were rated as significantly higher in arousal when paired with the high arousal label "distraught" ($M = 3.78$, $SD = 1.04$) than the low arousal label "down" ($M = 3.31$, $SD = 1.06$), $p = .007$. Angry faces paired with the high arousal label "furious" ($M = 4.83$, $SD = 1.01$) were rated as significantly more aroused than when paired with the low arousal label "grumpy" ($M = 3.91$, $SD = 1.19$), $p < .001$. Likewise, scared faces were rated higher in arousal when paired with the high arousal label "terrified" ($M = 5.22$, $SD = 1.13$) than when paired with the low arousal label "worried" ($M = 4.68$, $SD = 1.05$), $p < .001$. Finally, when paired with the high arousal label "victorious" ($M = 5.65$, $SD = 1.18$), proud faces were rated as more aroused than when paired with the low arousal label "satisfied" ($M = 5.19$, $SD = 0.89$), $p = .009$.

With regard to the supplemental Faces+Labels vs. Faces Alone analyses, we found that proud Faces+ high arousal Labels were rated as higher in arousal than proud Faces Alone ($p < .001$) (Supplemental Materials, S6 Appendix).

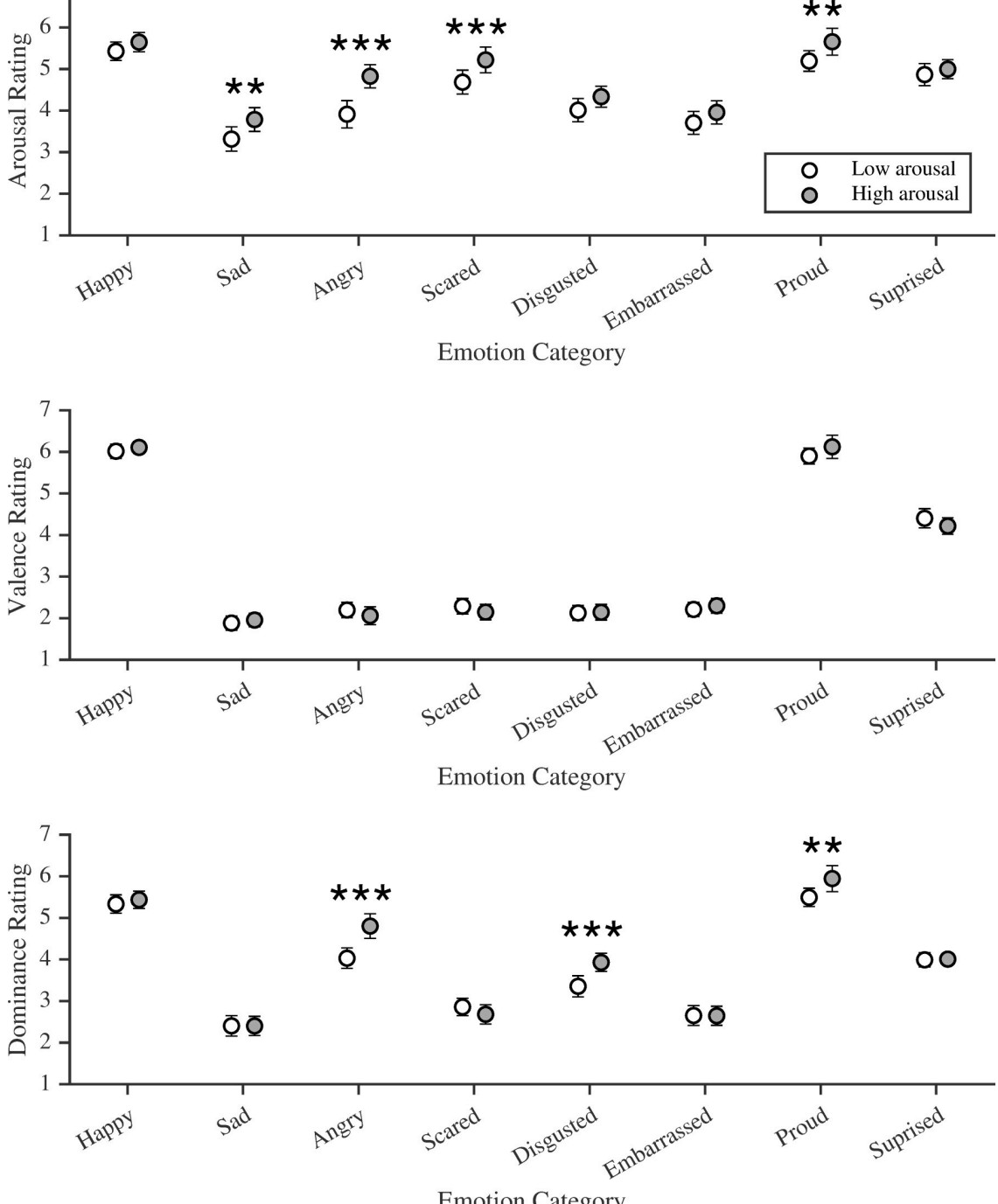

**Fig 1. Arousal, valence and dominance ratings for Faces+Labels (static stimuli).** Arousal (1 = *very sleepy*, to 7 = *very awake*), valence (1 = *very negative*, to 7 = *very positive*) and dominance (1 = *very weak*, to 7 = *very powerful*) ratings for static faces paired with low and high arousal labels, for each emotion category. Note: Bonferroni corrected * $p < .05$, ** $p < .01$, *** $p < .001$, two-tailed, 95% confidence intervals shown.

## Valence

As predicted, there was a significant main effect for emotion category, $F(2.73, 141.84) = 539.51$, $p < .001$, $\eta p2 = .912$, although no significant main effect of label arousal, $F(1, 52) = 0.01$, $p = .908$, $\eta p2 = 0$. There was a significant two-way interaction between emotion categories and label arousal, indicating that the effect of label arousal on valence ratings differed depending on the emotion category, $F(4.42, 230.05) = 2.65$, $p = .029$, $\eta p2 = .049$. However, no post hoc tests remained significant following Bonferroni correction, all $p$s = 1.00.

## Dominance

Dominance ratings for faces paired with labels also differed between emotion categories and label arousal level. There was a main effect for emotion category, $F(3.46, 179.82) = 174.77$, $p < .001$, $\eta p2 = .781$, a main effect for label arousal level, $F(1, 52) = 20.85$, $p < .001$, $\eta p2 = .286$, and a significant two-way interaction which indicated that the effect of label arousal on dominance ratings differed across emotion categories, $F(5.07, 263.48) = 12.07$, $p < .001$, $\eta p2 = .188$. This interaction was driven by responses to angry, disgusted, and proud faces (see Fig 1). Angry faces paired with the high arousal label "furious" ($M = 4.80$, $SD = 1.07$) were rated as higher in dominance than when paired with the low arousal label "grumpy" ($M = 4.03$, $SD = 0.89$), $p < .001$. Disgusted faces paired with the high arousal label "repulsed" ($M = 3.93$, $SD = 0.79$) were rated as more dominant than when presented with the low arousal label "nauseated" ($M = 3.35$, $SD = 0.92$), $p < .001$. When proud faces were paired with the high arousal label "victorious" ($M = 5.95$, $SD = 1.14$), they were rated as higher in dominance than when paired with low arousal label "satisfied" ($M = 5.49$, $SD = 0.80$), $p = .002$. These findings are in the same direction as the obtained dominance ratings for Labels Alone (see Supplemental Materials, S3 Appendix).

## Study 1 conclusion

Overall, in line with predictions, the results indicate that varying the arousal level of labels is associated with differences in how facial expressions are rated, in terms of arousal and dominance. Although the strength of this effect differs between emotion category and the labels used, these results suggest that even when the emotion label is unrelated to the goal of the task, participants are still influenced by the label provided.

## Study 2: Dynamic stimuli

Previous research looking at the influence of language on emotion judgments has found that when the face is less informative participants rely more on accompanying language information [41]. Study 1 used static expressions, which provide less information than dynamic expressions and are not representative of the expressions we see in daily communication [7,51]. Furthermore, research has indicated that dynamic expressions allow for better discrimination of ambiguous expressions than do static images [49,50,48,67]. The aim of Study 2 was to replicate Study 1 using dynamic stimuli, in order to rule out the possible explanation that the significant effects of labels on judgments of emotional expressions were due to the low information available in static faces.

## Method

**Participants.** One hundred and four first-year psychology students at the University of Queensland participated and received course credit. This sample consisted of 74 females and 29 males (one participant did not identify as either male or female), with a mean age of 25.46

years. Seventy-six participants reported that English was their primary language. All partici-
pants provided informed, written consent to participate.

## Materials

**Facial expressions.**   The same 32 expressions used in Study 1 were included in the current
study, but the expressions were presented as dynamic videos rather than static images. Each
video lasted approximately 5 seconds and captured an actor moving their face from a neutral
position to the apex of an expression. The videos were cut after the apex of the expression was
reached, and then disappeared from the screen. The videos were presented at a resolution of
720 x 576.

**Emotion word labels.**   The same set of 24 emotion labels from Study 1 was used in the cur-
rent study. Again, medium arousal labels were excluded from analyses.

## Dependent measures

As in Study 1, the dependent measures were perceived affect (i.e., valence, arousal and domi-
nance) of the faces and labels. Participants responded to each item using three 7-point, Likert-
type slider scales for arousal (1.00 = *very sleepy*, to 7.00 = *very awake*), valence (1.00 = *very neg-
ative*, to 7.00 = *very positive*), and dominance (1.00 = *very weak*, to 7.00 = *very powerful*), to
two decimal places. As per the method of Study 1, participants rated four expressions for each
emotion label, thus we averaged participants' ratings across faces for each emotion label.

## Procedure

The procedure was identical to that of Study 1, with the exception of the use of dynamic
expressions rather than static expressions. Participants were randomly assigned to view one of
two conditions: the Faces+Labels (*N* = 56) and Faces Alone (*N* = 48). Because the 24 labels
used in this study were identical to those of Study 1, we did not assign participants in this
study to a Labels Alone condition.

## Results

To investigate the influence of labels on judgments of emotional faces, we conducted an 8
(emotion category) x 2 (label arousal level) repeated-measures ANOVA for each dimension
measured (i.e., arousal, valence, dominance). Greenhouse-Geisser corrections were applied
when assumptions of sphericity were violated. As per Study 1, significant interactions were fol-
lowed up with post hoc paired-samples *t*-tests and Bonferroni-corrected *p* values are reported.
In the following sections we specify only the significant effects and contrasts. Faces+Labels vs.
Faces Alone comparisons are available in the Supplemental Material, S8 Appendix.

### Arousal

As in Study 1, arousal ratings for faces paired with labels differed across emotion categories
and label arousal level. There was a main effect of emotion category, $F(3.61, 198.72) = 69.17$,
$p < .001$, $\eta p2 = .557$, and a main effect of label arousal level, $F(1, 55) = 45.44$, $p < .001$ $\eta p2 =$
.452. A significant two-way interaction indicated that the effect of label arousal level on per-
ceived arousal from faces varied by emotion category, $F(4.99, 274.37) = 5.72$, $p < .001$ $\eta p2 =$
.094 (see Fig 2). Bonferroni-corrected post hoc analyses revealed a consistent pattern for sad,
angry and scared faces, where faces paired with labels higher in arousal were rated as higher in
arousal than those paired with lower arousal labels. Sad faces paired with high arousal label
"distraught" ($M = 3.62$, $SD = 1.01$) had higher arousal ratings than those paired with low

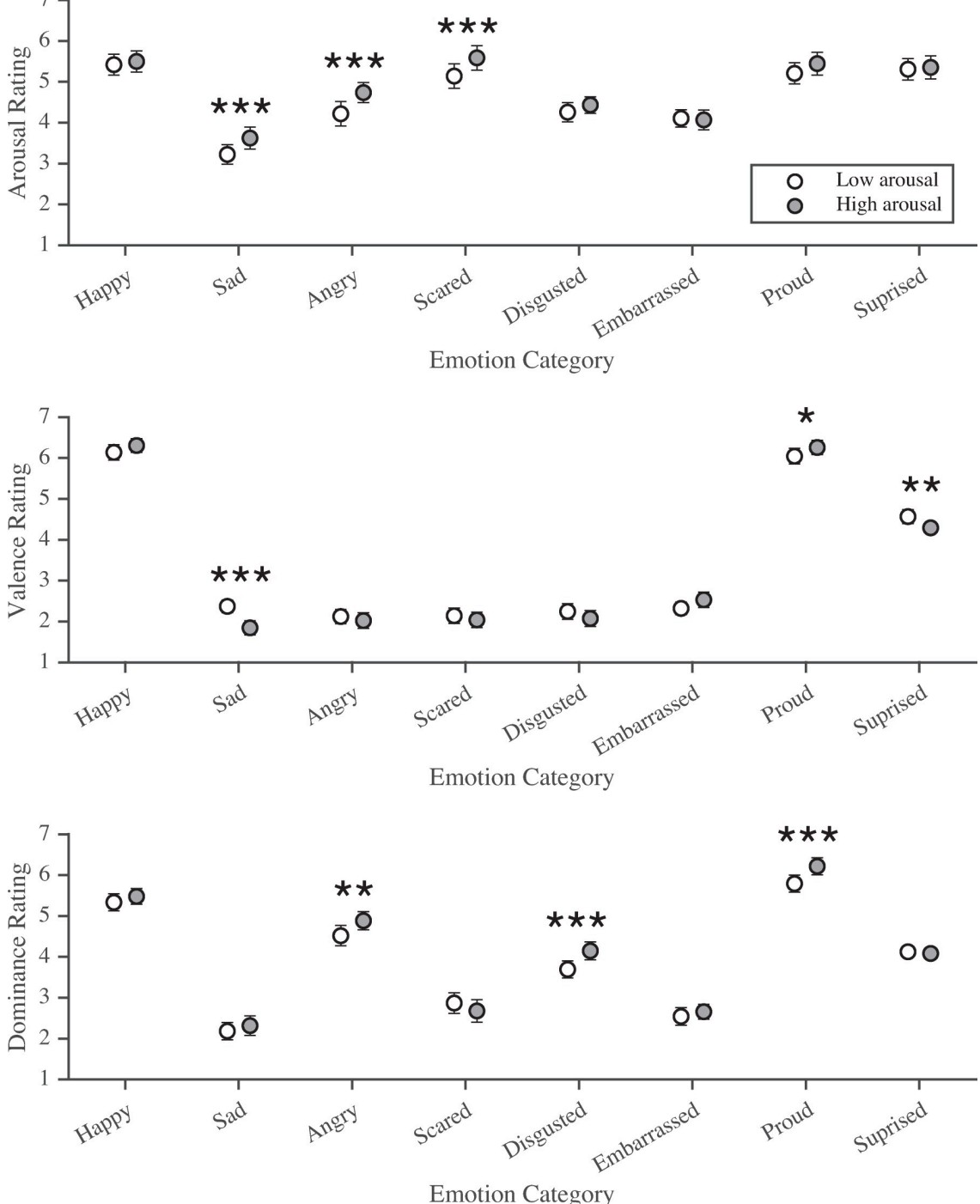

**Fig 2. Arousal, valence and dominance ratings for Faces+Labels (dynamic stimuli).** Arousal (1 = *very sleepy*, to 7 = *very awake*), valence (1 = *very negative*, to 7 = *very positive*) and dominance (1 = *very weak*, to 7 = *very powerful*) ratings for dynamic faces paired with low and high arousal labels, for each emotion category. Note: Bonferroni corrected * $p < .05$, ** $p < .01$, *** $p < .001$, two-tailed, 95% confidence intervals shown.

arousal label "down" ($M = 3.22$, $SD = 0.89$), $p < .001$. Angry faces paired with high arousal label "furious" ($M = 4.74$, $SD = 0.91$) were rated higher in arousal than those paired with low arousal label "grumpy" ($M = 4.22$, $SD = 1.12$), $p < .001$. Similarly, scared faces paired with high

arousal label "terrified" ($M$ = 5.59, $SD$ = 1.11) were rated higher in arousal than those paired with low arousal label "worried" ($M$ = 5.14, $SD$ = 1.11), $p$ < .001.

In terms of the supplemental Faces+Labels vs. Faces Alone analyses, we found that Faces + low arousal Labels were rated as lower in arousal than the Faces Alone for sad and angry faces ($p$s < .010) (Supplemental Materials, S8 Appendix).

## Valence

Valence ratings for faces paired with labels were different across emotion categories and label arousal level: there was a significant main effect for emotion category, $F(2.03, 101.72)$ = 507.72, $p$ < .001, ηp2 = .910, and a significant main effect for label arousal level, $F(1, 50)$ = 6.02, $p$ = .018, ηp2 = .107. A significant two-way interaction between emotion categories and label arousal indicated that the effect of label arousal on valence ratings differed depending on the emotion category, $F(5.62, 280.78)$ = 15.44, $p$ < .001, ηp2 = .236 (see Fig 2). For sad and surprised faces, more highly aroused labels were associated with more negative ratings, whereas lower arousal labels were rated as more positive. Sad faces paired with high arousal label "distraught" were rated as more negative ($M$ = 1.84, $SD$ = 0.62) than those paired with the low arousal label "down" ($M$ = 2.37, $SD$ = 0.57), $p$ < .001. Surprised faces paired with the high arousal label "astounded" ($M$ = 4.29, $SD$ = 0.51) were seen to be significantly more negative than faces paired with high label "awed" ($M$ = 4.57, $SD$ = 0.64), $p$ = .003. To the opposite effect, higher arousal labels were associated with more positive perceptions for proud faces; faces paired with high arousal label "victorious" ($M$ = 6.26, $SD$ = 0.65) were seen to be significantly more positive than those paired with low label "satisfied" ($M$ = 6.04, $SD$ = 0.72), $p$ = .040. These findings are in the same direction as the valence ratings for Labels Alone (see Supplemental Materials, S3 Appendix).

## Dominance

As in Study 1, dominance ratings for faces paired with labels were different across emotion categories and label arousal level. There was a main effect for emotion category, $F(3.63, 192.29)$ = 227.00, $p$ < .001, ηp2 = .811, and a main effect for label arousal level, $F(1, 53)$ = 30.63, $p$ < .001, ηp2 = .366. A significant two-way interaction indicated an effect of label arousal on perceived dominance of faces, depending on emotion category, $F(5.99, 317.57)$ = 6.70, $p$ < .001, ηp2 = .112. Specifically, angry, proud and disgusted faces were perceived as more dominant when paired with high arousal labels than when paired with low arousal labels (see Fig 2). Angry faces paired with high arousal label "furious" ($M$ = 4.88, $SD$ = 0.83) were rated as more dominant than for low arousal "grumpy" ($M$ = 4.52, $SD$ = 0.93), $p$ = .002. Proud faces paired with high arousal label "victorious" ($M$ = 6.22, $SD$ = 0.78) were perceived as significantly more dominant than those paired with low arousal label "satisfied" ($M$ = 5.79, $SD$ = 0.78), $p$ < .001. Disgusted faces paired with high arousal label "repulsed" ($M$ = 4.15, $SD$ = 0.82) were perceived to be significantly more dominant than those paired with low arousal label "nauseated" ($M$ = 3.69, $SD$ = 0.78), $p$ < .001. These findings are in the same direction as the dominance ratings of Labels Alone (see Supplemental Materials, S3 Appendix).

## Study 2 conclusion

Replicating the findings of Study 1, these results indicate that faces are rated differently in terms of arousal, valence and dominance depending on the arousal level of the associated labels. Again, the strength and direction of this effect was dependent on the emotion category and the labels used. Using dynamic stimuli did not eliminate the influence of language on

facial expression judgments. Direct static vs. dynamic comparisons are available in the Supplemental Materials, S9 Appendix.

## Discussion

There is a growing body of research providing evidence for the role of contextual information in emotion perception (for a review see 39). We investigated the effect of emotion labels on the perceived arousal, valence, and dominance of facial expressions, using both static and dynamic stimuli. Overall, our results suggest that the emotion labels accompanying facial expressions influence emotion judgments of those expressions, although this was true for only some emotion categories in the current studies. Our findings are consistent with previous studies showing that labels influence emotion perception [41,47].

### Perceived arousal

We have demonstrated that labels conveying the same emotion category but differing in levels of arousal (e.g. "worried" vs. "terrified" for the category of fear), impact participants' ratings of facial expressions. In both studies, labels that were higher in arousal led to judgments of the accompanying sad, angry, and scared expressions as displaying higher arousal than when those same expressions were accompanied by a low arousal label.

**The influence of motion on arousal ratings.** Although ratings of most expressions were similar across Studies 1 and 2, our findings regarding proud expressions differed across the two studies. When static expressions were presented (Study 1), high arousal labels prompted participants to judge faces as higher in arousal, but this effect was not found when dynamic expressions were used (Study 2). Previous research has shown that when pride expressions are presented dynamically, participants require less expression information to accurately judge the expression [51]. It is possible that when presented with dynamic pride expressions in this study, participants gathered sufficient arousal information from the expression and therefore arousal judgments were less influenced by the label information. However, this finding should be replicated with additional research.

### Perceived valence

Variations in label arousal also influenced participants' ratings of valence for dynamic expressions only. In line with previous research [e.g., 60], the direction of this influence varied with the emotion category presented, with higher arousal labels prompting participant judgments of the accompanying facial expressions toward the extreme ends of the scale and away from neutrality. In particular, higher arousal labels prompted participants to judge the accompanying dynamic proud expressions as more positive in valence, whereas sad and surprised expressions were judged as more negative. Importantly, these findings were in the same direction the valence ratings for the labels.

**The influence of motion on valence ratings.** Our results suggest that labels influenced valence ratings of dynamic expressions but not static expressions, although why this occurs is unclear. Previous research has demonstrated that when expressions of pride are presented dynamically, participants are more sensitive to variations in expression valence [48]; that is, participants detect valence differences in dynamic proud expressions but not static expressions. Unlike arousal, where dynamic expressions potentially created *less ambiguity* resulting in less label influence, it is possible that, in terms of valence, dynamic expressions actually created room for an influence of labels, which was not available for static expressions. However, it is important to note that this is speculation based only on the pride literature; it remains

unknown whether similar effects are evident across other emotions such as surprise and sadness. This is a topic that should be addressed in future research.

## Perceived dominance

Variations in label arousal also influenced participants' ratings of dominance in expressions. In both studies, labels that were higher in arousal prompted participants to judge the accompanying angry, proud, and disgust expressions as displaying higher dominance than those same expressions were accompanied by a low arousal label. Expressions of anger and pride have been associated with aggression and status [20,48,68,69] and high arousal labels may have amplified this association for participants.

Consistent with the dominance ratings the Labels Alone condition, disgusted faces paired with high arousal label "repulsed" were judged as more dominant than faces paired with low arousal label "nauseated" in both studies. It is possible that pairing the label "nauseated" with disgusted faces may have suggested to participants a physical disgust, and the label "repulsed" may be more representative of moral disgust [70,71]. Thus, it is possible that when disgusted faces were paired with labels associated with highly aroused, moral disgust, participants associated these traits with increased dominance in the expressions.

## Theoretical implications

Our results stand in line with a constructionist theory of emotion, which posits that language information provides context that allows us to perceive emotion from facial expressions [7]. Here we extend the previous work to show that emotion labels not only shift the emotion concepts perceived in a facial expression, but they shift the underlying affective dimensions as well [39–41]. It is possible that these dimensional judgments shift because the labels influence the extent to which underlying dimensional concepts are activated, thus, prompting the participant to judge the face to be at a different place on the affective circumplex.

The slightly differing pattern of results between static and dynamic expressions may also have theoretical implications. Although dynamic expressions have more information inherent in them than static expressions [49–51], the current results tentatively suggest that label information has a qualitatively stronger effect on judgments of dynamic expressions (at least for valence), which is somewhat counter-intuitive. This highlights the importance of considering motion as a factor influencing affective judgments, and the complexity of how contextual information is integrated. However, these results are preliminary and need replication before theoretical mechanisms can be speculated.

## Limitations and future directions

Of the eight expression categories we presented, six were influenced by the contextual information added by emotion labels, with four emotion categories influenced in terms of arousal ratings (sad, angry, scared, proud). However, label information did not appear to be integrated into ratings of happy and embarrassed expressions. One explanation is that judgments of happy and embarrassed faces were not influenced by the language information for these emotions. For embarrassed faces, it is also possible that the difference in arousal between "ashamed" vs. "mortified" was not great enough to elicit effects of label arousal on judgments of the face. Alternatively, it is possible that judgments of happy and embarrassed expressions did change, but on an unmeasured dimension (e.g., trustworthiness) [72]. Future research using a broader array of dimensions could address this possibility. Likewise, future studies should include additional emotion categories, such as contempt.

Future research should also consider employing a within-subjects design, whereby all participants rate Faces Alone, Labels Alone and Faces+Labels, as this would help to minimise variability that arises from between-subjects designs. In the current study, Faces Alone vs. Faces+Labels comparisons did not reveal strong differences (see Supplemental Materials, Appendices F, H); such effects are likely subtle and require within-person methods to detect. Moreover, the addition of a control condition that includes mismatched words and faces (e.g., angry face paired with label "contented") would help disentangle whether the effects of the label on face judgments are related to priming [see 73] or the perceiver combining or integrating language and face information to make affective judgments.

In this research we systematically varied emotion word arousal; however, emotion words also varied in valence and dominance (see Supplemental Materials, S3 Appendix). This highlights the complexity of language as a form of context, in that it provides information to perceivers through multiple modes and influences affective perception accordingly. Future research should include more than one label per level of arousal in each emotion category, as well as more extreme labels, and investigate how valence and dominance ratings shift accordingly. It is also possible that by rating the three dimensions at the same time, participants' responses were influenced by their ratings of the other dimensions–future research might consider having participants rate each dimension in separate blocks. Replicating our study methodology using a different and larger selection of labels would be important to establish generalisability of results, and would provide insight into the broader influence of language on perceived expression affect. Future research could also design a paradigm that would allow us to specifically measure unique contributions of face information and label information to arousal / valence / dominance judgments. Similarly, future research should investigate the influence of faces on label ratings, to gain a more complete understanding of the potential bidirectional relationship between language and emotional expressions. We expect that label ratings would be influenced by paired face information; similar bidirectional relationships have previously been shown between face-body and face-voice pairings [74–80], and we believe this would hold for face-language pairings.

Finally, the expressions used in the current studies represent highly caricatured expressions of emotion. The use of more realistic and less extreme expression stimuli in future studies would further our understanding of how language context is integrated into emotion judgments during real-world emotion perception. It could be that linguistic contextual information plays an even greater role in influencing our emotion judgments during daily interactions.

## Conclusion

Previous research has shown that judgments of discrete emotion categories are influenced by the presentation of language context [12,24,25,42]. The current studies demonstrate that language context alters the dimensional affective foundations that underlie our judgments of others' expressions. Emotions are not simply read from expressions. Rather, language may have the last word.

## Supporting information

**S1 Appendix. Pilot data.**
(DOCX)

**S2 Appendix. Example of test trial format.**
(DOCX)

**S3 Appendix. Means and standard deviations of arousal, valence and dominance ratings for low and high arousal labels alone.**
(DOCX)

**S4 Appendix. Means and standard deviations of all label arousal ratings (incl. medium arousal labels).**
(DOCX)

**S5 Appendix. Main analyses including medium arousal labels (static stimuli).**
(DOCX)

**S6 Appendix. Faces+Labels vs. Faces alone (static stimuli).**
(DOCX)

**S7 Appendix. Main analyses including medium arousal labels (dynamic stimuli).**
(DOCX)

**S8 Appendix. Faces+Labels vs. Faces alone (dynamic stimuli).**
(DOCX)

**S9 Appendix. Dynamic vs static stimuli.**
(DOCX)

**S10 Appendix. Diamond plots showing all raw data for Faces+Labels ratings.**
(DOCX)

## Acknowledgments

We thank Jo Butler, Trudy McCaul, Teresa Nguyen and Tamara Van Der Zant for their assistance with data collection, and Luke Hearne for his assistance with figures.

## Author Contributions

**Conceptualization:** Megan S. Barker, Nicole L. Nelson.

**Data curation:** Megan S. Barker, Emma M. Bidstrup.

**Formal analysis:** Megan S. Barker, Emma M. Bidstrup, Nicole L. Nelson.

**Methodology:** Megan S. Barker, Emma M. Bidstrup.

**Project administration:** Emma M. Bidstrup.

**Resources:** Nicole L. Nelson.

**Supervision:** Gail A. Robinson, Nicole L. Nelson.

**Writing – original draft:** Megan S. Barker, Emma M. Bidstrup.

**Writing – review & editing:** Megan S. Barker, Gail A. Robinson, Nicole L. Nelson.

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
