## [Decision Letter · Decision Letter 0]

5 Dec 2019

PONE-D-19-26857

"Irritated" or "Furious"? Arousal of Emotion Labels Influences Judgments of Facial Expressions

PLOS ONE

Dear Dr. Barker,

Thank you for submitting your manuscript to PLOS ONE. After careful consideration, we feel that it has merit but does not fully meet PLOS ONE’s publication criteria as it currently stands. Therefore, we invite you to submit a revised version of the manuscript that addresses the points raised during the review process.

Although the reviewers were generally enthusiastic about the paper, they also raised serious concerns about the methodology and implications of the findings.  

We would appreciate receiving your revised manuscript by Jan 19 2020 11:59PM. To enhance the reproducibility of your results, we recommend that if applicable you deposit your laboratory protocols in protocols.io, where a protocol can be assigned its own identifier (DOI) such that it can be cited independently in the future. For instructions see: http://journals.plos.org/plosone/s/submission-guidelines#loc-laboratory-protocols

We look forward to receiving your revised manuscript.

Kind regards,

Peter A. Bos

Academic Editor

PLOS ONE

Journal Requirements:

Additional Editor Comments (if provided):

Reviewers' comments:

Reviewer's Responses to Questions

**Comments to the Author**

1. Is the manuscript technically sound, and do the data support the conclusions?

Reviewer #1: Partly

Reviewer #2: Partly

Reviewer #3: Yes

2. Has the statistical analysis been performed appropriately and rigorously? 

Reviewer #1: Yes

Reviewer #2: Yes

Reviewer #3: Yes

3. Have the authors made all data underlying the findings in their manuscript fully available?

Reviewer #1: Yes

Reviewer #2: Yes

Reviewer #3: Yes

4. Is the manuscript presented in an intelligible fashion and written in standard English?

Reviewer #1: Yes

Reviewer #2: Yes

Reviewer #3: Yes

5. Review Comments to the Author

Reviewer #1: The present manuscript investigates the potential effects of linguistic context information on emotion perception. Here, faces displaying different emotional expressions were paired with different emotion labels which expressed either a high or a low arousal state and each face+label combination was rated for arousal, valence and dominance (the three main dimensions according to the dimensional account of emotion perception and recognition). What is good and novel about the paper is the additional manipulation of face stimulus presentation (static vs dynamic) which strengthens the generalisability of the results and gets us a bit closer to understanding emotion perception in the real world. While the main question of the paper is interesting and definitely worth investigating, I find the results and their interpretation very underwhelming. The authors show that if we vary the arousal level of the emotion labels, this would change the overall perception of emotional arousal for some but not all emotional expressions. It is really not clear how these findings might have important implications for what we already know about emotion perception. I would encourage the authors to run an additional study designed to specifically estimate how much of the overall arousal perception is due to the emotion label and how much of it is due to the face information available. This can be easily done by manipulating both the arousal levels of the label (high vs low) as well as the arousal levels shown in the faces. This will increase the interest and the theoretical contribution of the paper.

Please find a list of my major and more minor points for each section of the manuscript below:

Introduction

- It might be helpful to provide a more balanced discussion of the two accounts of emotion perception. Also, from the authors' description it sounds like both accounts rely on signals from the face – it is not clear how the findings of this study support one account over the other. There is no doubt that other contextual information and language, in particular, could affect emotion perception – it is just not clear how this relates and helps to compare the two models of emotion perception.

- p. 7 – The authors try to justify why their study was focused specifically on the arousal dimension – it would be worth supporting their arguments with some research evidence.

- Up until the last paragraph in the introduction, nothing has been mentioned on the difference between emotion perception from static and dynamic stimuli – it will be helpful to include some addition material earlier on in the text.

- The authors do not make their hypotheses regarding the different emotional expressions as well as the different emotion ratings (arousal, valence and dominance) clear in the text.

Method

- How was sample size determined?

- p. 9 – “The posers varied across emotion categories” – This is a bit unclear: there are 4 identities and 8 emotions which makes a total of 32 stimuli. Doesn’t that mean that every poser was pictured with all emotional expressions?

- The labels of the arousal scale seem very irrelevant and ambiguous for an emotion perception task. It is difficult to imagine how the word worried or ashamed can be described as either sleepy or awake.

- It would be helpful to provide a statistical comparison between the three label categories (high, moderate and low) using data from the pilot study. This could also be included as Supplementary Material.

- In the procedure section the authors say that participants were explicitly asked to provide ratings based on the face information only. This is an important point that should be addressed throughout the text. It changes the main question of the study from how emotional labels change emotion perception to whether emotional labels and face information are automatically integrated together. This could lead to an alternative interpretation of the results – if participants’ ratings are influenced by the emotion labels then they were most likely automatically integrated with the face and participants were unable to ignore them. If participants’ ratings were not influenced by the labels then this implies that labels and faces were not integrated together.

- There are significant differences between sample sizes across the three conditions (e.g., 53 participants in the face+labels condition and 71 participants in the faces only condition). Same was true for the number of trials in each condition. Authors should provide some explanation for these differences.

- How were the stimuli rated for arousal, valence and dominance? Were all dimensions rated at the same time or in separate blocks? If they were all rated together – how did the authors minimise carryover effects?

Results

- An important issue in any scale rating is participants’ agreement. Therefore, it might be helpful to include some measure of rater agreement in the beginning of the Results section.

- Ratings of labels only – The results in this section are presented rather selectively (last paragraph p. 10). The authors do not report all results across all emotional expressions and arousal conditions (e.g., only results for high but not low arousal labels are reported in relation to perceived dominance).

- Valence results – first sentence, p. 15 – The authors state that valence ratings differed across label arousal conditions but this is not reflected in their graph or in their statistical analysis.

- It would be helpful to include a graph illustrating the comparison between static and dynamic stimuli. Also, the authors need to report both main effects throughout the text.

Discussion

- The discussion leaves many open questions that the authors should address:

- Why do arousal labels influence arousal perception of faces for only certain emotions?

- Why do we see differences in the patterns of results from label only ratings and face+labels ratings?

- Why arousal labels did not seem to influence valence ratings for static stimuli?

- How do these findings relate to the two emotion accounts mentioned in the introduction?

- Why do we see differences in the patterns of results with static and dynamic face presentation?

- The authors state that the results from the face+labels condition fit well with ratings from the labels only condition but this point needs further elaboration. For example, we see significant differences for sadness in valence ratings and while it is true that the labels only ratings show low valence ratings for the high arousal label, such differences exist for most other emotions, yet we did not see significant differences with these emotions in terms of valence. This needs to be further addressed in the discussion.

- I would encourage the authors to soften their claims in the discussion. Their findings do not really show that some emotion categories are more susceptible to the influences of labels than others but rather that some are influenced, and some are not. Also, the statement that their predicted effects work for negative emotions is also not entirely valid as there are other negative emotions (such as disgust or embarrassment) that were not influenced by the labels. They will need to provide some possible explanation why this would be the case.

- The Future Directions section begins with a slightly misleading statement. The main focus of the paper is arousal ratings and only half of the emotions were influenced by the emotion labels.

- The authors argue that they do not find any significant effects for happiness and embarrassment because participants were not influenced by the labels for these emotions, but this is not reflected in the ratings shown in Appendix A.

- Looking at the analyses provided in the Supplementary Materials regarding the comparison between the face only and the face+label conditions, raises some serious concerns about the validity of the main results and the authors’ interpretation. These analyses show that for the most part there were no differences in the ratings provided in these two conditions which implies that the labels might not have had a considerable effect on the overall emotion ratings. This is an important point and should be addressed in the main text of the paper.

Reviewer #2: Across two studies, the authors show that emotion labels shift perceptions of the affect expressed by both static (Study 1) and dynamic (Study 2) expressions. Although results are highly consistent across studies, some interesting discrepancies arise, which the authors carefully discuss. I’ll cut to the chase and say that I really like this paper. It uses a simple and clear methodology to test an interesting and important question in affective science. There’s a growing body of work showing how language influences emotion experience and perception, and I think this paper uses a clever design to make an important contribution to this body of research. I can’t wait to cite it. I also really appreciated how clearly the article was written and that the authors were measured about the strengths and limitations of their study. Below I provide both larger and smaller comments that I hope are helpful to the authors as they polish up this really interesting paper.

Large comments

- The authors show that the high arousal and low arousal words do not differ on arousal alone (they also vary on valence and dominance), but they frame their paper as being about how “high arousal” and “low arousal” labels shift emotion perception throughout the title/abstract/main text. This is a bit of a logical problem, because it’s not just the arousal that differs between these sets of words. I don’t think this is a problem with the study itself, as the authors still show that language activates concepts, which shapes perception (an interesting finding), but their current decision to focus only on the arousal of the words leads them to make claims that are a little confounded. They’ll need to reframe this slightly. For example, they might need to say up front that you’re using “arousal” as shorthand for “an overall more activating/extreme/dominant label, which varies on all these dimensions”, or they’ll need to use some other word to refer to the two different sets of labels.

- I understand the authors’ desire to compare results from Study 1 and Study 2, but I just don’t think that can be done… The two studies were collected separately, meaning that participants weren’t randomly assigned to the dynamic v static conditions. This opens up a huge number of personal and temporal confounds that render any comparison of the two studies impossible. I think it’s fine to compare qualitative differences across the study (e.g., labels affected perceptions of some emotions for static but not dynamic stimuli), but any statistical comparisons aren’t really allowable given that the samples are dissimilar. I think the entire section of the results comparing the studies needs to be removed, all comparisions must be purely qualitative, and randomly assigning Ps to conditions can be discussed as a future direction.

- I was somewhat confused by the authors’ decision to frame the emotion debate as between “basic” and “dimensional” accounts and to only bring up constructionism midway through the introduction. If there’s a compelling reason for this, I’d love to hear it explained, but personally I feel like the authors’ data are just fantastically in line with the constructionist model and could be better articulated within this framework itself. In fact, I found myself wanting a little more of a theoretical conversation about their data in the discussion. How exactly do they think that language shifts perceptions of affect? They set up this logic in the introduction, but providing a clearer “account” in the discussion would be helpful. Additionally, I’d recommend tweaking the introduction to jump straight into constructionist theory (without stopping at the dimensional account) and then shift the conclusions to be within the constructionist model (dimensional perceptions shift because language influences what concepts people use to understand expressions, and these concepts are associated with different places on the circumplex). This would also affect the conclusion in the abstract. If the authors instead want to argue from a slightly different theoretical framing, that’s totally fine, but being clearer about how they see the psychological process occurring would be helpful to readers.

- There are some recent citations that I think would really help the authors support key points they make in the introduction: emotion expressions don’t reliably communicate specific emotion types (Barrett, Adolphs, Marsella, Martinez, & Pollak, 2019), language serves as a context for emotion (Lindquist, Satpute, & Gendron, 2015), language development shapes emotion development (Nook, Sasse, Lambert, McLaughlin, & Somerville, 2017; Hoemann, Xu, & Barrett, in press); priming emotion concepts influences emotion perception (Fugate, Gouzoules, Barrett, 2010; Halberstadt, 2003, 2005; Halberstadt & Niedenthal, 2001; Nook, Lindquist, & Zaki, 2015; Plate, Wood, Woodard, & Pollak, in press; Satpute et al., 2016, PsychSci). This last group of articles in particular could use greater review in the introduction, given that the methods and focus of the work is so overlapping.

- Additional information regarding the studies’ methods should be provided. Let us know enough detail so we could run the study ourselves (e.g., timing of all components of the trial, were responses self-paced or under a time limit, how long did dynamic expressions last, what was the intertrial stimulus & interval, what software did you use to run the study, etc.)

- The studies have healthy sample sizes, but no power analyses are reported. If they weren’t conducted a priori, that’s ok, but providing a post-hoc power analysis (and being transparent about it) would be helpful in letting readers know how well powered the sample is to find effects of a given size.

- A final concern readers might have is that the finding is all “just priming”. Now, I think that a process of priming (language –> concept –> perception) is actually occurring in a meaningful sense, but one concern that lingers in this design is that labels presented with any face will shift perceptions of that face. In other words, merely seeing the label “irritated” makes people use higher ends of the scale compared to seeing “furious.” Ideally, there would be a control condition to handle this, where labels were paired with truly mismatched faces (e.g., calm, which comes from the opposite quadrant of the circumplex) so you could show that what really matters is that the conceptual label combines with affective information presented in the face to produce emotion perception. (In this way, the current study is a really nice perceptual equivalent of the Lindquist & Barrett, 2008 PsychSci paper). Not having this control condition isn’t damning to the paper, but I think it should be noted as a limitation and future direction.

Small comments

- In the introduction, the header “Studies 1 and 2” is a little confusing (I thought you were suddenly out of the introduction). Perhaps change to “The Current Study” or a description of the hypotheses you are testing.

- I was surprised that happy faces were seen as so arousing overall. Any thoughts on that? Any possibility this is because the likert poles were labeled with “sleepy” and “awake”, and happy faces just seemed the most awake rather than the most “activated”?

- Do you have any sense if participants were aware that the same exact stimulus was said to be expressing multiple different emotions? Do you have any thoughts on how this might have influenced results?

- In the discussion, you say that “the systematic manipulation of language information according to valence, dominance or other dimensions would provide valuable insight into the influence of language on perceived expression affect” but I don’t know if such a thing would even be possible. These underlying dimensions seem to share some structure (as you point out in the U-shaped relations between them). I’d avoid dwelling on the idea that these dimensions need to be cleanly separated, because it seems like affect doesn’t really work that way and instead focus on how cool it is that words shift affective perceptions.

- I love the idea to investigate whether the effect works in the opposite direction (i.e., pairing faces that differ in affective intensity with the same label to see how faces shift affective perceptions of words). I’d spell this out a little more and say what you’d hypothesize.

- Adding figures in the SupMat describing results of those analyses would be helpful for readers.

- It looks like comparisons between face+label and face only conditions didn’t reveal strong differences overall. This is interesting (and I don’t think problematic), so I’d recommend mentioning and briefly discussing that result in the main text. It is likely that these linguistic effects effects require within-person methods to detect. The papers I list above also speak to the power of within-person designs.

Reviewer #3: Dear Authors,

I have had the opportunity to review your submission, “Irritated or Furious? Arousal of Emotion Labels Influences Judgments of Facial Expressions”. This paper addresses an interesting question about the role of emotion words on the perception of facial depictions of emotion, a topic that is well-aligned and largely overlapping with studies that I have performed myself.

Strengths:

1. This is a very straight-forward experimental design in which the condition of interest has participants receiving 8 different emotions with an arousal-congruent (high or low) label and indicating ratings of arousal, valence, and dominance.

2. The background and, for the most part, the philosophical grounding is clear. My only suggestion would be to entertain more than the 2 most disparate theories of emotion: basic and dimensional. I would recommend briefly also reviewing psychological (e.g. Barrett) and social constructionism (e.g. Russell), the former which is touched upon later.

3. Writing is clear and easily followed.

4. The logic of the studies are straight-forward, well-executed and analyzed. I have done very similar studies and find the methods and analyses to be in alignment.

5. The hypotheses are clear and straight-forward.

Suggestions for publication:

1. As is currently stands, I think this paper makes a small contribution to the field and is worth publishing, but the effect of emotion words on judgments using facial expressions has been tested several times now and a mechanism is even speculated (e.g. Barrett, 2017b; Clark, 2013; Hutchinson & Barrett, 2019;Lupyan & Clark, 2015; Lupyan & Ward, 2013). To this end, Fugate and colleagues (Fugate, O’Hare, & Emmanuel, 2018) tested the effect of emotion words on the perception of moving emotional faces, not so different from the dynamic presentation here in Study 2. I would ask that the researchers look at this work, as those authors suggest that congruent emotion words (compared to control words) decrease category variability among examplars. In a similar study to the current, the same first author also showed that labeling facial expressions with partial or fully congruent emotion words (including on arousal, but also on valence) shifts participants’ perception of facial expressions to align with the word (Fugate, Gendron, et al., 2018, Experiment 3). I think both could be helpful to discuss in the introduction and to the current authors in furthering their research. Therefore, I think the impact of the current results is limited (although fully supported).

I would like to see a replication as a within-subjects design, in which the three conditions (face, face + label, label) could be compared directly. Of course, there might be carry-over effects from one condition to the next, but simply counterbalancing the order should take care of that. This would provide more compelling evidence of how labels (generally) compared to no-labels affect perception.

I would also like (or in place of the first suggestion) to see the study repeated with different (equally-arousing) words so that we can generalize the findings to beyond the specific words chosen.

Minor Criticisms/Suggestions/Clarifications:

In addition to the suggestions noted above, there are few minor (easily addressed) things to address:

1. Just to be clear, it seems that not all 8 emotions per ID are used, but rather that each emotion is shown with four different IDs. I think that is fine, but please be clear that is the case. Moreover, were there any differences among IDs for an emotion in any of the ratings? It would be good to check.

2. It seems that the range of arousal (between the high and low words) is not great, and I wonder whether there would be more/different effects if higher average “high” arousal and lower average “low” arousal words could be found/used. To that end, I would like to see that the range between high and low words for an emotion is similar across emotions. It seems that they might differ, and this would need to be addressed if so.

3. I’d like to see a correlation run between arousal ratings and dominance ratings for at least study 1. It seems the two are highly correlated. If so, I’d like to see an explanation of why the authors treated dominance as a separate dimension.

4. The authors don’t really discuss the fact that arousal words seem to affect valence ratings but only for the dynamic faces. Although I agree that the pattern between Study 1 and 2 is largely similar, I think why valence is disproportionately affected in Study 2 is worth expanding upon.

5. The information in Table 1 is repeated in Appendix A, and therefore I recommend just replacing Table 1 with the Appendix table.

6. Overall, the discussion seems largely a summary of the results and therefore overlaps the direct comparison of study 1 with 2. I recommend that the discussion try to address some broader issues rather than re-summarize what was already nicely summarized.

Overall, I think this paper is easily revised to take into account the minor criticisms. Without additional data collection, however, I am unsure how much a contribution this article makes. My perception of PLoS ONE is that it publishes articles which are typically more groundbreaking or at least impactful than the systematic progression of this line of research, but I will leave that decision up to the editor.

Thank you for the time to review this paper.

Kind Regards,

6. PLOS authors have the option to publish the peer review history of their article (what does this mean?). If published, this will include your full peer review and any attached files.

Reviewer #1: No

Reviewer #2: Yes: Erik Nook

Reviewer #3: Yes: Jennifer MB Fugate

---

## [Author Response · Author response to Decision Letter 0]

19 May 2020

Dear Prof. Bos,

Re: PONE-D-19-26857

‘Grumpy’ or ‘Furious’? Arousal of Emotion Labels Influences Judgments of Facial Expressions (Barker, Bidstrup, Robinson and Nelson)

Thank you for the positive and helpful comments on our manuscript and inviting us to submit a revised version to PLOS ONE. All three reviewers commented that it was an interesting study, but expressed concerns and suggestions for us to incorporate in a revision. For your convenience, we reproduce the reviewers’ comments verbatim with our responses positioned under each comment. Changes made to the manuscript are in italic text.

The Editor requested upon revision that we specify (1) whether consent was informed and (2) what type you obtained (for instance, written or verbal, and if verbal, how it was documented and witnessed). We have responded to this by adding the following to the main text of the manuscript: “All participants provided informed, written consent to participate.” (p. 8, p.17)

Reviewer #1: 

1. The present manuscript investigates the potential effects of linguistic context information on emotion perception. Here, faces displaying different emotional expressions were paired with different emotion labels which expressed either a high or a low arousal state and each face+label combination was rated for arousal, valence and dominance (the three main dimensions according to the dimensional account of emotion perception and recognition). What is good and novel about the paper is the additional manipulation of face stimulus presentation (static vs dynamic) which strengthens the generalisability of the results and gets us a bit closer to understanding emotion perception in the real world. While the main question of the paper is interesting and definitely worth investigating, I find the results and their interpretation very underwhelming. The authors show that if we vary the arousal level of the emotion labels, this would change the overall perception of emotional arousal for some but not all emotional expressions. It is really not clear how these findings might have important implications for what we already know about emotion perception. I would encourage the authors to run an additional study designed to specifically estimate how much of the overall arousal perception is due to the emotion label and how much of it is due to the face information available. This can be easily done by manipulating both the arousal levels of the label (high vs low) as well as the arousal levels shown in the faces. This will increase the interest and the theoretical contribution of the paper.

• We appreciate the suggestion to collect additional data, but unfortunately we are not in a position to run another study. However, we have added this reviewer’s recommendation as a direction for future research in the Discussion (p. 26): “Future research could also design a paradigm that would allow us to specifically measure unique contributions of face information and label information to arousal / valence / dominance judgments.”

Introduction

2. It might be helpful to provide a more balanced discussion of the two accounts of emotion perception. Also, from the authors' description it sounds like both accounts rely on signals from the face – it is not clear how the findings of this study support one account over the other. There is no doubt that other contextual information and language, in particular, could affect emotion perception – it is just not clear how this relates and helps to compare the two models of emotion perception.

• Regarding the “Perspectives on Emotion” paragraph in the Introduction: our intention was not to for our study to lend support to the basic emotion approach vs. dimensional approach, and we agree that this was confusing. For this reason, we have restructured the Introduction to bring constructionism in earlier, as this is the theoretical framework that best fits our study (p. 4) (see also Reviewer 2 Major Point 3). 

3. p. 7 – The authors try to justify why their study was focused specifically on the arousal dimension – it would be worth supporting their arguments with some research evidence.

• We appreciate the opportunity to clarify our thinking here. We had drawn most of our words from the ANEW database (Bradley & Lang, 1999). In this database, the standard deviation of valence ratings was smaller than that of the arousal ratings (1.6 vs. 2.2 on a 7-point Likert scale). Given we had elected to vary our words on only one dimension, these numbers prompted us to manipulate arousal. We now make this clearer in the paper, stating: “We opted to manipulate arousal in emotion labels (rather than valence) because the large database of participant word ratings we planned to use (59) showed that arousal ratings were more variable across words than valence ratings.” (p. 7)

4. Up until the last paragraph in the introduction, nothing has been mentioned on the difference between emotion perception from static and dynamic stimuli – it will be helpful to include some addition material earlier on in the text.

• We now provide additional information about dynamic vs. static stimuli in the introduction, stating: “The influence of movement in judging others’ expressions can also provide important context, and better approximate the moving expressions observed in daily life (51). Previous research has shown that dynamic expressions provide more information than static expressions and influence how features of expressions are visually processed.” (p. 6). 

5. The authors do not make their hypotheses regarding the different emotional expressions as well as the different emotion ratings (arousal, valence and dominance) clear in the text.

• We appreciate this point but we did not have clear hypotheses regarding different emotion categories a priori; rather, we were interested in looking at broad patterns in this exploratory research. We now make this explicit on pg. 6, stating: “To ensure we examined broad patterns in judgements of facial expressions, we incorporated a variety of facial expressions.” 

• We highlight in the Introduction that the arousal, valence and dominance ratings will be influenced in line with the label ratings for these dimensions (p. 8): “Moreover, we hypothesised that label arousal level would also influence the valence and dominance ratings of faces consistent with the U-shaped curve seen in previous research”

Method

6. How was sample size determined?

• Our goal was to collect approximately 50 participants per condition, with the stopping point being the end of the academic semester. This sample size based on standard practice in the field. For example, recent studies by Fugate and colleagues, which use similar methodology, include sample sizes between 30 and 50 (Fugate, Gendron, et al., 2018; Fugate, O’Hare, et al., 2018). Our Faces+Labels condition is similar to this: N = 53 (Study 1) and N = 56 (Study 2). We now make this clear on page 8 under “Participants”.

7. p. 9 – “The posers varied across emotion categories” – This is a bit unclear: there are 4 identities and 8 emotions which makes a total of 32 stimuli. Doesn’t that mean that every poser was pictured with all emotional expressions?

• Thank you for allowing us the opportunity to clarify this point, which was also raised by Reviewer 3. For each of the eight emotions four identities were shown, but these were not always the same posers. We have changed the text to make this point clearer (p. 9): “We selected four expressions (two male posers and two female posers) for each of the eight expressions included: happy, sad, angry, scared, disgusted, embarrassed, proud and surprised. The four expressions for each emotion were drawn from posers within the ADFES set, with some posers repeating across emotion categories, but no poser displayed expressions for all emotions included.” 

8. The labels of the arousal scale seem very irrelevant and ambiguous for an emotion perception task. It is difficult to imagine how the word worried or ashamed can be described as either sleepy or awake.

• These labels are fairly standard, and have been used in previous research by other researchers (e.g. Russell, 1980; Russell & Paris, 1994; Widen & Russell, 2016); as well as in our lab (e.g. Vidas et al, 2020). 

9. It would be helpful to provide a statistical comparison between the three label categories (high, moderate and low) using data from the pilot study. This could also be included as Supplementary Material.

• As requested by this reviewer, all pilot data and t-test comparisons between low, moderate and high label arousal ratings are now available in the Supplemental Material, and the reader is directed to this on p. 10. 

• We highlight that the purpose of the pilot was to select labels to use in the main studies, and we conducted a label manipulation check which is reported in the main text. 

10. In the procedure section the authors say that participants were explicitly asked to provide ratings based on the face information only. This is an important point that should be addressed throughout the text. It changes the main question of the study from how emotional labels change emotion perception to whether emotional labels and face information are automatically integrated together. This could lead to an alternative interpretation of the results – if participants’ ratings are influenced by the emotion labels then they were most likely automatically integrated with the face and participants were unable to ignore them. If participants’ ratings were not influenced by the labels then this implies that labels and faces were not integrated together.

• This study was indeed about the integration of face and language information, and our interpretation of the results is that emotion labels were integrated with the face. In fact, our primary aim stated in the Introduction is: “The current studies aim to investigate the influence of emotion word labels, on the judgment of emotion in facial expressions… here we investigate whether linking facial expressions with emotion labels of varying arousal levels will influence participants’ affective judgments of facial expressions.” (p. 6-7)

11. There are significant differences between sample sizes across the three conditions (e.g., 53 participants in the face+labels condition and 71 participants in the faces only condition). Same was true for the number of trials in each condition. Authors should provide some explanation for these differences.

• Our goal was to collect approximately 50 participants per condition across the three conditions, although for one condition (Static Faces Alone) we had access to ~20 additional participants. If this reviewer would like us to balance the sample sizes by excluding participants then we can do so, but we would prefer not to exclude data on this basis. 

12. How were the stimuli rated for arousal, valence and dominance? Were all dimensions rated at the same time or in separate blocks? If they were all rated together – how did the authors minimise carryover effects?

• All dimensions were rated at the same time, following the presentation of the face. To minimize participant confusion, and increase the accuracy of the data collected, the dimensions were always presented on a single page, in the same order: valence, arousal, dominance. We have added this detail into the Procedure (p. 10). It is possible that there were carryover effects, but the anchors of the scales were all available and the three dimensions are separable. An example of the test trial format is now available in the Supplemental Materials. 

Results

13. An important issue in any scale rating is participants’ agreement. Therefore, it might be helpful to include some measure of rater agreement in the beginning of the Results section.

• We appreciate that rater agreement is important. To address this reviewer’s concern, we have added standard deviations where means are reported (p. 13-16; 19-22), which give the reader an indication of the variability in ratings. Furthermore, we have added diamond plots depicting all the raw data into the Supplemental Material; these show the entire range of participant responses for all stimuli. Finally, we draw the reviewer’s attention to the narrow error bars in Figures 1 and 2, which indicate consistency in participants’ ratings. 

14. Ratings of labels only – The results in this section are presented rather selectively (last paragraph p. 10). The authors do not report all results across all emotional expressions and arousal conditions (e.g., only results for high but not low arousal labels are reported in relation to perceived dominance).

• We believe the reviewer has misunderstood this analysis. This reviewer gives the example that we “only [report] results for high but not low arousal labels in relation to perceived dominance”. In fact, the t-test comparison was between high and low arousal labels, and thus the test statistic reflects the difference between high and low. To clarify this point we now begin the sentences with: “compared to low arousal labels, high arousal…” (e.g. p. 11).

• We had also previously reported only the emotion categories that differed significantly in terms of valence and dominance in the Labels Manipulation Check (ratings of labels alone). To address this reviewer’s concern we have now also added the non-significant post-hoc results into the main text (p. 11): “High vs. low arousal labels for “happiness” “anger” “embarrassment” and “pride” were not significantly different in terms of valence, all ps > .05.”; “High vs. low arousal labels for “happiness” “sadness” and “fear” were not significantly different in terms of dominance, all ps > .05.”

15. Valence results – first sentence, p. 15 – The authors state that valence ratings differed across label arousal conditions but this is not reflected in their graph or in their statistical analysis.

• Although there were significant main effects of emotion category and label arousal, the post hoc t-tests did not remain significant following stringent Bonferroni correction. To address this reviewer’s concern we have removed the first sentence from the Study 1 Valence Results paragraph (p. 16). 

16. It would be helpful to include a graph illustrating the comparison between static and dynamic stimuli. Also, the authors need to report both main effects throughout the text.

• In response to Reviewer 2 Major Point 2, we have opted to remove this analysis from the main text of the paper, and now include it in the Supplemental Material. This is because the static vs. dynamic statistical comparison is highly controversial, considering that the data for the two studies were collected separately and therefore participants were not randomly assigned to conditions (static vs. dynamic). 

• However, as requested, we now include figures illustrating the static vs. dynamic comparison alongside the analyses in the Supplemental Material.

Discussion

17. The discussion leaves many open questions that the authors should address:

a) Why do arousal labels influence arousal perception of faces for only certain emotions?

b) Why do we see differences in the patterns of results from label only ratings and face+labels ratings?

c) Why arousal labels did not seem to influence valence ratings for static stimuli?

d) How do these findings relate to the two emotion accounts mentioned in the introduction?

e) Why do we see differences in the patterns of results with static and dynamic face presentation?

f) The authors state that the results from the face+labels condition fit well with ratings from the labels only condition but this point needs further elaboration. For example, we see significant differences for sadness in valence ratings and while it is true that the labels only ratings show low valence ratings for the high arousal label, such differences exist for most other emotions, yet we did not see significant differences with these emotions in terms of valence. This needs to be further addressed in the discussion.

• This reviewer raises some thought-provoking questions. However, we would like to highlight that this research was largely exploratory and we do not feel comfortable speculating post hoc, as this would be poor research practice. This is especially true when considering explanations about specific emotion categories (e.g., why valence ratings for sadness but not other emotions differed – questions [a] and [f]). As we point out in the Discussion, our results are limited to the set of emotion labels we employed, and therefore we cannot draw conclusions about specific emotion categories without additional research. What we can highlight in these data are broad patterns suggesting that judgments of facial expressions vary depending on the paired language information. However, we have expanded our Discussion to focus more on emotion theories (p. 25) and we hope that this addresses any outstanding questions this reviewer may have. 

• Question [b]: We expected the Labels Alone ratings and the Faces+Labels ratings to differ, because in one condition only one source of emotional information is considered and in the Faces+Labels condition two sources are – the Faces+Labels condition was designed to see how these two sources of information would be integrated. We have further explained this in the discussion – we hope this version is clearer. 

• Question [c]: This point was also raised by Reviewer 3. To address this question, we have added the following into the Discussion (p. 24): “Previous research has demonstrated that when expressions of pride are presented dynamically, participants are more sensitive to variations in expression valence (51); that is, participants detect valence differences in dynamic proud expressions but not static expressions. Unlike arousal, where dynamic expressions potentially created less ambiguity resulting in less label influence, it is possible that, in terms of valence, dynamic expressions actually created room for an influence of labels, which was not there for static expressions. However, it is important to note that this is speculation based only on the pride literature; it remains unknown whether similar effects are evident across other emotions such as surprise and sadness.”

• Question [d]: As per our response to Point 2, we were not intending to provide support for the basic emotion approach vs. dimensional approach, and have restructured the Introduction to make our theoretical grounding in constructionism clear. 

18. I would encourage the authors to soften their claims in the discussion. Their findings do not really show that some emotion categories are more susceptible to the influences of labels than others but rather that some are influenced, and some are not. 

• We have re-phrased our statement in the first paragraph of the Discussion; instead of stating that some emotion categories are more susceptible to the influence of labels, we now say (p. 22): “Overall, our results suggest that the emotion labels accompanying facial expressions influence emotion judgments of those expressions, although this was true for only some emotion categories in the current studies.”

19. Also, the statement that their predicted effects work for negative emotions is also not entirely valid as there are other negative emotions (such as disgust or embarrassment) that were not influenced by the labels. They will need to provide some possible explanation why this would be the case.

• To address this reviewer’s concern we have removed the sentence stating that perceptions of arousal in faces are more easily manipulated by language information for negative emotions (p. 23), rather than trying to post-hoc rationalize why some negative emotions did not show the same pattern as others in our data. 

20. The Future Directions section begins with a slightly misleading statement. The main focus of the paper is arousal ratings and only half of the emotions were influenced by the emotion labels.

• We disagree that this statement is misleading, as it is true that six of the eight emotion categories were influenced by language information on some dimension (arousal, valence or dominance). However, we have now added context for clarification (p. 25): “Of the eight expression categories we presented, six were influenced by the contextual information added by emotion labels, with four emotion categories influenced in terms of arousal ratings (sad, angry, scared, proud).”

21. The authors argue that they do not find any significant effects for happiness and embarrassment because participants were not influenced by the labels for these emotions, but this is not reflected in the ratings shown in Appendix A.

• This reviewer is correct that we suggest that judgments of happy and embarrassed faces were not influenced by the language information. We based this statement on the fact that Faces+Labels ratings did not change depending on the paired labels for these two emotions. We believe this reviewer has misread Appendix A, as these ratings are for labels only (without faces); while we see variability in ratings of “contented” vs. “elated” (happiness) and “ashamed” vs. “mortified” (embarrassment) when the labels are presented alone, judgments of faces paired with these labels were uninfluenced by the label information. This is reflected in Figures 1 and 2. 

22. Looking at the analyses provided in the Supplementary Materials regarding the comparison between the face only and the face+label conditions, raises some serious concerns about the validity of the main results and the authors’ interpretation. These analyses show that for the most part there were no differences in the ratings provided in these two conditions which implies that the labels might not have had a considerable effect on the overall emotion ratings. This is an important point and should be addressed in the main text of the paper.

• We now include new figures in the Supplemental Materials (Figure SM2, SM4), which better show the variability between the Faces+Labels and Faces Alone ratings. Our primary interest was how face ratings varied across label conditions, with the comparison between Faces Alone and Face+Label as a secondary analysis. It is likely that these effects are subtle and require within-person paradigms to detect, and we have added this as a limitation and direction for future research (p. 25-26): “Future research should also consider employing a within-subjects design, whereby all participants rate Faces Alone, Labels Alone and Faces+Labels, as this would help to minimise variability that arises from between-subjects designs. In the current study, Faces Alone vs. Faces+Labels comparisons did not reveal strong differences (see Supplemental Materials); such effects are likely subtle and require within-person methods to detect.”

Reviewer #2: 

Across two studies, the authors show that emotion labels shift perceptions of the affect expressed by both static (Study 1) and dynamic (Study 2) expressions. Although results are highly consistent across studies, some interesting discrepancies arise, which the authors carefully discuss. I’ll cut to the chase and say that I really like this paper. It uses a simple and clear methodology to test an interesting and important question in affective science. There’s a growing body of work showing how language influences emotion experience and perception, and I think this paper uses a clever design to make an important contribution to this body of research. I can’t wait to cite it. I also really appreciated how clearly the article was written and that the authors were measured about the strengths and limitations of their study. Below I provide both larger and smaller comments that I hope are helpful to the authors as they polish up this really interesting paper.

Major Comments

1. The authors show that the high arousal and low arousal words do not differ on arousal alone (they also vary on valence and dominance), but they frame their paper as being about how “high arousal” and “low arousal” labels shift emotion perception throughout the title/abstract/main text. This is a bit of a logical problem, because it’s not just the arousal that differs between these sets of words. I don’t think this is a problem with the study itself, as the authors still show that language activates concepts, which shapes perception (an interesting finding), but their current decision to focus only on the arousal of the words leads them to make claims that are a little confounded. They’ll need to reframe this slightly. For example, they might need to say up front that you’re using “arousal” as shorthand for “an overall more activating/extreme/dominant label, which varies on all these dimensions”, or they’ll need to use some other word to refer to the two different sets of labels.

• Thank you for this comment. Arousal is the dimension we manipulated systematically, which is why we focus on the arousal of the words. We certainly appreciate that valence and dominance shift as arousal changes, and indeed this was our primary motivation for gathering valence and dominance ratings of Faces+Labels alongside arousal ratings. 

• However, we are not comfortable with the idea of using “arousal” as shorthand for an overall more activating/extreme/dominant label, because the relationship between these three dimensions is not linear. To address this reviewer’s concern we do now highlight in the introduction that while we systematically varied arousal, manipulating that variable also influences other dimensions (p. 7-8)

2. I understand the authors’ desire to compare results from Study 1 and Study 2, but I just don’t think that can be done… The two studies were collected separately, meaning that participants weren’t randomly assigned to the dynamic v static conditions. This opens up a huge number of personal and temporal confounds that render any comparison of the two studies impossible. I think it’s fine to compare qualitative differences across the study (e.g., labels affected perceptions of some emotions for static but not dynamic stimuli), but any statistical comparisons aren’t really allowable given that the samples are dissimilar. I think the entire section of the results comparing the studies needs to be removed, all comparisons must be purely qualitative, and randomly assigning Ps to conditions can be discussed as a future direction.

• Thank you for raising this important point. We agree with the reviewer, and understand that it is controversial; therefore we have removed it from the main text into the Supplemental Material. However, we have opted to retain it in the Supplemental Material because this is a question that we have been asked to address multiple times by different reviewers (e.g. Reviewer 1 Point 16). 

3. I was somewhat confused by the authors’ decision to frame the emotion debate as between “basic” and “dimensional” accounts and to only bring up constructionism midway through the introduction. If there’s a compelling reason for this, I’d love to hear it explained, but personally I feel like the authors’ data are just fantastically in line with the constructionist model and could be better articulated within this framework itself. In fact, I found myself wanting a little more of a theoretical conversation about their data in the discussion. How exactly do they think that language shifts perceptions of affect? They set up this logic in the introduction, but providing a clearer “account” in the discussion would be helpful. Additionally, I’d recommend tweaking the introduction to jump straight into constructionist theory (without stopping at the dimensional account) and then shift the conclusions to be within the constructionist model (dimensional perceptions shift because language influences what concepts people use to understand expressions, and these concepts are associated with different places on the circumplex). This would also affect the conclusion in the abstract. If the authors instead want to argue from a slightly different theoretical framing, that’s totally fine, but being clearer about how they see the psychological process occurring would be helpful to readers.

• Our intention was not to frame the emotion debate as between basic and dimensional accounts in the Introduction, although we understand how it came across this way. We have restructured and rewritten parts of the Introduction to make clear that our study rationale is grounded in constructionist theory, in line with this reviewer’s recommendation (p. 4-5) (see also Reviewer 1 Major Point 2). 

• As requested, have now include a “Theoretical Implications” section in the Discussion (p. 25)

4. There are some recent citations that I think would really help the authors support key points they make in the introduction: emotion expressions don’t reliably communicate specific emotion types (Barrett, Adolphs, Marsella, Martinez, & Pollak, 2019), language serves as a context for emotion (Lindquist, Satpute, & Gendron, 2015), language development shapes emotion development (Nook, Sasse, Lambert, McLaughlin, & Somerville, 2017; Hoemann, Xu, & Barrett, in press); priming emotion concepts influences emotion perception (Fugate, Gouzoules, Barrett, 2010; Halberstadt, 2003, 2005; Halberstadt & Niedenthal, 2001; Nook, Lindquist, & Zaki, 2015; Plate, Wood, Woodard, & Pollak, in press; Satpute et al., 2016, PsychSci). This last group of articles in particular could use greater review in the introduction, given that the methods and focus of the work is so overlapping.

• Thank you for drawing our attention to these articles; we have included this literature in the Introduction and we believe that our rationale is now clearer. 

5. Additional information regarding the studies’ methods should be provided. Let us know enough detail so we could run the study ourselves (e.g., timing of all components of the trial, were responses self-paced or under a time limit, how long did dynamic expressions last, what was the intertrial stimulus & interval, what software did you use to run the study, etc.)

• As suggested, we have now included additional detail about the methods in the manuscript: “The experiment was run using Qualtrics, and responses were self-paced with participants allowed to take as long as they liked to answer the questions” (p. 10) and “Each video lasted approximately 5 seconds and captured an actor moving their face from a neutral position to the apex of an expression.” (p. 18). 

• In addition, now include an example of a test trial in the Supplemental Materials.

6. The studies have healthy sample sizes, but no power analyses are reported. If they weren’t conducted a priori, that’s ok, but providing a post-hoc power analysis (and being transparent about it) would be helpful in letting readers know how well powered the sample is to find effects of a given size.

• Thank you for this suggestion. We now include the following text in the Supplemental Materials: “A post-hoc power analysis in G*Power (Faul, Erdfelder, Buchner, & Lang, 2009) found that in order to detect a small effect (Cohen's f = .10; α = .05; β = .80) for a 2 (condition) x 8 (emotion) repeated measures ANOVA, a sample of 92 participants (46 per group) was required. We have exceeded this number.”

7. A final concern readers might have is that the finding is all “just priming”. Now, I think that a process of priming (language –> concept –> perception) is actually occurring in a meaningful sense, but one concern that lingers in this design is that labels presented with any face will shift perceptions of that face. In other words, merely seeing the label “irritated” makes people use higher ends of the scale compared to seeing “furious.” Ideally, there would be a control condition to handle this, where labels were paired with truly mismatched faces (e.g., calm, which comes from the opposite quadrant of the circumplex) so you could show that what really matters is that the conceptual label combines with affective information presented in the face to produce emotion perception. (In this way, the current study is a really nice perceptual equivalent of the Lindquist & Barrett, 2008 PsychSci paper). Not having this control condition isn’t damning to the paper, but I think it should be noted as a limitation and future direction.

• The effect of priming is an interesting possibility that we now raise in the Discussion as a future direction (p. 26): “The addition of a control condition that includes mismatched words and faces (e.g., angry face paired with label “contented”) would help disentangle whether the effects of the label on face judgments are related to priming or the perceiver combining language and face information to make affective judgments.”

Minor Comments

1. In the introduction, the header “Studies 1 and 2” is a little confusing (I thought you were suddenly out of the introduction). Perhaps change to “The Current Study” or a description of the hypotheses you are testing.

• As suggested, the heading of this section has been changed to “The Current Studies” (p. 6).

2. I was surprised that happy faces were seen as so arousing overall. Any thoughts on that? Any possibility this is because the likert poles were labeled with “sleepy” and “awake”, and happy faces just seemed the most awake rather than the most “activated”?

• We note that the faces rated in the current studies were posed, highly caricatured expressions. These expressions are known to be particularly intense, which is part of what we think participants were rating. We acknowledge that highly intensified happy faces to not bear exact resemblance to realistic expressions of happiness, and we note in the Discussion that using more realistic and less extreme stimuli would be important in furthering our understanding of how the emotion-language relationship operates in the real world (p. 26-27). 

3. Do you have any sense if participants were aware that the same exact stimulus was said to be expressing multiple different emotions? Do you have any thoughts on how this might have influenced results?

• We were not concerned about participants potentially being aware that the same face was paired with multiple labels from an emotion category because: 1) we still saw variability in Face+Label judgments between label arousal levels, and 2) if participants were aware that the same face was being shown multiple times, we would expect less variability in judgments of the faces (i.e., this effect would be working in opposition to our hypothesis). We anticipate that if there was a unique actor for every Face+Label combination then the results would actually be more extreme. Therefore, we believe that participants seeing each face paired with multiple labels is a strength of our experimental design. 

4. In the discussion, you say that “the systematic manipulation of language information according to valence, dominance or other dimensions would provide valuable insight into the influence of language on perceived expression affect” but I don’t know if such a thing would even be possible. These underlying dimensions seem to share some structure (as you point out in the U-shaped relations between them). I’d avoid dwelling on the idea that these dimensions need to be cleanly separated, because it seems like affect doesn’t really work that way and instead focus on how cool it is that words shift affective perceptions.

• We appreciate the point that whether we can clearly dissociate arousal, valence and dominance is not entirely clear. As suggested, we have removed the sentence quoted by the reviewer (p. 26). 

5. I love the idea to investigate whether the effect works in the opposite direction (i.e., pairing faces that differ in affective intensity with the same label to see how faces shift affective perceptions of words). I’d spell this out a little more and say what you’d hypothesize.

• We agree that this potential bidirectional relationship is of great interest to future research, and now make our predictions clear in the Discussion (p. 26): “We expect that label ratings would be influenced by paired face information; similar bidirectional relationships have previously been shown between face-body and face-voice pairings, and we believe this would hold for face-language pairings.”

6. Adding figures in the SupMat describing results of those analyses would be helpful for readers.

• As requested, we have added figures showing the results of all Supplementary analyses (Figures SM1 to SM4). 

7. It looks like comparisons between face+label and face only conditions didn’t reveal strong differences overall. This is interesting (and I don’t think problematic), so I’d recommend mentioning and briefly discussing that result in the main text. It is likely that these linguistic effects require within-person methods to detect. The papers I list above also speak to the power of within-person designs.

• We agree that these results are difficult to interpret because of the between-subjects design, and that a within-subjects experiment might be needed to detect Faces Alone vs. Faces+Labels differences. We now highlight this in the main text (p. 25-26): “Future research should also consider employing a within-subjects design, whereby all participants rate Faces Alone, Labels Alone and Faces+Labels, as this would help to minimise variability that arises from between-subjects designs. In the current study, Faces Alone vs. Faces+Labels comparisons did not reveal strong differences (see Supplemental Materials); such effects are likely subtle and require within-person methods to detect.”

Reviewer #3: 

I have had the opportunity to review your submission, “Irritated or Furious? Arousal of Emotion Labels Influences Judgments of Facial Expressions”. This paper addresses an interesting question about the role of emotion words on the perception of facial depictions of emotion, a topic that is well-aligned and largely overlapping with studies that I have performed myself.

Strengths:

1. This is a very straight-forward experimental design in which the condition of interest has participants receiving 8 different emotions with an arousal-congruent (high or low) label and indicating ratings of arousal, valence, and dominance.

2. The background and, for the most part, the philosophical grounding is clear. My only suggestion would be to entertain more than the 2 most disparate theories of emotion: basic and dimensional. I would recommend briefly also reviewing psychological (e.g. Barrett) and social constructionism (e.g. Russell), the former which is touched upon later.

• Thank you for this comment. We have now restructured the Introduction to present constructionism upfront, as this theoretical framework provided the basis for our study (see also Reviewer 1 Major Point 2 and Reviewer 2 Major Point 3). In addition, we now include more detail in our description of constructionism. While we can appreciate the differences between psychological and social constructionism, we believe that this level of detail is beyond the scope of the current paper.

3. Writing is clear and easily followed.

4. The logic of the studies are straight-forward, well-executed and analyzed. I have done very similar studies and find the methods and analyses to be in alignment.

5. The hypotheses are clear and straight-forward.

• We thank this reviewer for the positive feedback and for highlighting the strengths of our paper. 

Suggestions for publication:

1. As is currently stands, I think this paper makes a small contribution to the field and is worth publishing, but the effect of emotion words on judgments using facial expressions has been tested several times now and a mechanism is even speculated (e.g. Barrett, 2017b; Clark, 2013; Hutchinson & Barrett, 2019;Lupyan & Clark, 2015; Lupyan & Ward, 2013). To this end, Fugate and colleagues (Fugate, O’Hare, & Emmanuel, 2018) tested the effect of emotion words on the perception of moving emotional faces, not so different from the dynamic presentation here in Study 2. I would ask that the researchers look at this work, as those authors suggest that congruent emotion words (compared to control words) decrease category variability among examplars. In a similar study to the current, the same first author also showed that labeling facial expressions with partial or fully congruent emotion words (including on arousal, but also on valence) shifts participants’ perception of facial expressions to align with the word (Fugate, Gendron, et al., 2018, Experiment 3). I think both could be helpful to discuss in the introduction and to the current authors in furthering their research. Therefore, I think the impact of the current results is limited (although fully supported).

• Thank you for suggesting the inclusion of these recently published studies. We now present the recent studies by Fugate and colleagues in the Introduction (p. 6): “Recently, Fugate et al. reported that participants were faster to detect a change in a transitioning face (e.g., moving from anger to disgust) when an emotion word congruent with the first emotion was presented, and slower when an incongruent word was presented. Similarly, the same research group showed that presenting emotion words following a target face shifted perception of the face (that is subsequently selecting the target face from an array) to align with the word, especially when exposure to the target face was limited.” 

• We also cite these studies in the Discussion (p. 23).

2. I would like to see a replication as a within-subjects design, in which the three conditions (face, face + label, label) could be compared directly. Of course, there might be carry-over effects from one condition to the next, but simply counterbalancing the order should take care of that. This would provide more compelling evidence of how labels (generally) compared to no-labels affect perception.

• We understand the power and increased experimental control of within-subjects designs, and while we are not in a position to run an additional study, we have added this point as a future direction (p. 25): “Future research should also consider employing a within-subjects design, whereby all participants rate Faces Alone, Labels Alone and Faces+Labels, as this would help to minimise variability that arises from between-subjects designs.” 

3. I would also like (or in place of the first suggestion) to see the study repeated with different (equally-arousing) words so that we can generalize the findings to beyond the specific words chosen.

• This is an important suggestion that has been added to our directions for future research (p. 26): “Replicating our study methodology using a different and larger selection of labels would be important to establish generalisability of results.”

Minor Comments:

1. Just to be clear, it seems that not all 8 emotions per ID are used, but rather that each emotion is shown with four different IDs. I think that is fine, but please be clear that is the case. Moreover, were there any differences among IDs for an emotion in any of the ratings? It would be good to check.

• This point has been clarified in text (p. 9): “We selected four expressions (two male posers and two female posers) for each of the eight expressions included: happy, sad, angry, scared, disgusted, embarrassed, proud and surprised. The four expressions for each emotion were drawn from posers within the ADFES set, with some posers repeating across emotion categories, but no poser displayed expressions for all emotions included.”

• We do not have access to individual ratings of posers, but we now point out that “The ADFES expressions are standardized and coded using the Facial Action Coding System. In a validation study, all expressions in the set were identified as the intended emotion by participants in 74% of trials” (footnote p. 9).

2. It seems that the range of arousal (between the high and low words) is not great, and I wonder whether there would be more/different effects if higher average “high” arousal and lower average “low” arousal words could be found/used. To that end, I would like to see that the range between high and low words for an emotion is similar across emotions. It seems that they might differ, and this would need to be addressed if so.

• We agree that a larger range of labels should be used in future research, and have addressed this in the Discussion (p. 26) “Future research should consider including more than one label per level of arousal in each emotion category, as well as more extreme labels”. We anticipate that the effect sizes would be larger if higher “high” and lower “low” arousal words could be found.

• To address this reviewer’s concern that the difference in arousal between high and low arousal words was similar for each emotion category, we have computed the differences and now include this information in Appendix A. The majority of emotions were similar in terms of arousal ratings: happy, sad, scared, disgusted, proud were all 1.1-1.5 points different. There was a greater difference between high and low arousal words for angry (2.6 points), while embarrassed and surprised had less than 1 point difference between high and low arousal words. The effect that these differences between emotions may have had on the results is now addressed in the Discussion (p. 25): “For embarrassed faces, it is also possible that the difference in arousal between ‘ashamed’ vs. ‘mortified’ was not great enough to elicit effects of label arousal on judgments of the face.”

3. I’d like to see a correlation run between arousal ratings and dominance ratings for at least study 1. It seems the two are highly correlated. If so, I’d like to see an explanation of why the authors treated dominance as a separate dimension.

• As requested, we have run Pearson’s correlations between arousal and dominance ratings for Study 1. We opted to run these correlations for each emotion separately, because while arousal and dominance are likely to track together for some emotions (e.g. anger), other emotions become lower in dominance as arousal increases (e.g. fear).

• The results are as follows:

Happiness: r = 0.57 p <.001

Sadness: r = 0.28 p <.001

Anger: r = 0.44 p <.001

Fear: r = -0.05 p =.546

Disgust: r = 0.28 p <.001

Embarrassment: r = 0.10 p = .194

Pride: r = 0.56 p <.001

Surprise: r = 0.15 p =.057

• These correlations show that arousal and dominance were not significantly correlated for all emotion categories (fear, embarrassment, surprise). We have opted to keep treating them as separate categories because it is standard in the field to use these three dimensions of affect. Also, we do not believe treating them as separate dimensions affects any conclusions drawn from the study, as we emphasise multiple times that the dimensions are related and influence one another. 

4. The authors don’t really discuss the fact that arousal words seem to affect valence ratings but only for the dynamic faces. Although I agree that the pattern between Study 1 and 2 is largely similar, I think why valence is disproportionately affected in Study 2 is worth expanding upon.

• As suggested, we have added a brief discussion about why we think valence might be disproportionately influenced in Study 2 (p. 24): “Previous research has demonstrated that when expressions of pride are presented dynamically, participants are more sensitive to variations in expression valence (51); that is, participants detect valence differences in dynamic proud expressions but not static expressions. Unlike arousal, where dynamic expressions potentially created less ambiguity resulting in less label influence, it is possible that, in terms of valence, dynamic expressions actually created room for an influence of labels, which was not there for static expressions. However, it is important to note that this is speculation based only on the pride literature; it remains unknown whether similar effects are evident across other emotions such as surprise and sadness.”

5. The information in Table 1 is repeated in Appendix A, and therefore I recommend just replacing Table 1 with the Appendix table.

• We have considered this reviewer’s suggestion; however, we believe that including all the information available in Appendix A in the main paper confuses the central point, which is that we systematically varied the arousal levels of the labels and not valence / dominance. Furthermore, Appendix A has been revised to address the point raised by this reviewer (Minor Point 2), and now includes differences in arousal / valance / dominance ratings between high and low arousal words, which adds even more information that is peripheral to the message of the paper. Therefore, for now we have left Table 1 as is, but we are happy to switch it with Appendix A if the editor wishes. 

6. Overall, the discussion seems largely a summary of the results and therefore overlaps the direct comparison of study 1 with 2. I recommend that the discussion try to address some broader issues rather than re-summarize what was already nicely summarized.

• As recommended by this reviewer, we have broadened our Discussion to address theoretical implications of our findings (p. 25). 

7. Overall, I think this paper is easily revised to take into account the minor criticisms. Without additional data collection, however, I am unsure how much a contribution this article makes. My perception of PLoS ONE is that it publishes articles which are typically more groundbreaking or at least impactful than the systematic progression of this line of research, but I will leave that decision up to the editor.

• We agree that this paper is easily revised and thank the reviewer for the helpful suggestions. However, while we appreciate the point about impact, we draw this reviewer’s attention to the mission statement of PLoS ONE: “We evaluate submitted manuscripts on the basis of methodological rigor and high ethical standards, regardless of perceived novelty.” https://journals.plos.org/plosone/s/journal-information

We thank the reviewers for their insightful comments and believe that these revisions have strengthened the study. 

Sincerely,

Megan Barker, Ph.D. (on behalf of co-authors)

---

## [Decision Letter · Decision Letter 1]

8 Jun 2020

PONE-D-19-26857R1

"Grumpy" or "Furious"? Arousal of Emotion Labels Influences Judgments of Facial Expressions

PLOS ONE

Dear Dr. Barker,

Thank you for submitting your manuscript to PLOS ONE. After careful consideration, we feel that it has merit but does not fully meet PLOS ONE’s publication criteria as it currently stands. Therefore, we invite you to submit a revised version of the manuscript that addresses the points raised during the review process.

Two of the reviewers were pleased with the revision, but one reviewer has some remaining concerns. I assume that these comments can be addressed by the authors in a final revision round. 

We look forward to receiving your revised manuscript.

Kind regards,

Peter A. Bos

Academic Editor

PLOS ONE

Reviewers' comments:

Reviewer's Responses to Questions

**Comments to the Author**

1. If the authors have adequately addressed your comments raised in a previous round of review and you feel that this manuscript is now acceptable for publication, you may indicate that here to bypass the “Comments to the Author” section, enter your conflict of interest statement in the “Confidential to Editor” section, and submit your "Accept" recommendation.

Reviewer #1: (No Response)

Reviewer #2: All comments have been addressed

Reviewer #3: All comments have been addressed

2. Is the manuscript technically sound, and do the data support the conclusions?

Reviewer #1: Yes

Reviewer #2: Yes

Reviewer #3: Yes

3. Has the statistical analysis been performed appropriately and rigorously? 

Reviewer #1: Yes

Reviewer #2: Yes

Reviewer #3: Yes

4. Have the authors made all data underlying the findings in their manuscript fully available?

Reviewer #1: Yes

Reviewer #2: Yes

Reviewer #3: Yes

5. Is the manuscript presented in an intelligible fashion and written in standard English?

Reviewer #1: Yes

Reviewer #2: Yes

Reviewer #3: Yes

6. Review Comments to the Author

Reviewer #1: I would like to thank the authors for their careful response to all of my many comments and for providing additional analyses in the Supplementary Materials. Overall, I think they have addressed the majority of my concerns although there are still a few outstanding issues. Please find a detailed list below:

1. This is not a concern, but I would like to mention that the introduction reads much better now and it motivates the two studies nicely.

2. I appreciate that the authors have provided a hypothesis regarding their predicted U-shaped curve describing the relationship between arousal, valence and dominance. It might be helpful to clarify exactly how they expect this to be demonstrated in the data from the present two studies. Looking at Figure 1, it seems like some of the results might not fit such a relationship – this should be discussed later on in the text.

3. I’m not entirely sure the power analysis provided by the authors is correct. A post hoc analysis provides information about the achieved power, not about the sample size needed to achieve that power. Moreover, since they are using a within-subjects design (2 conditions x 8 emotions), there should be only one group of participants.

4. In their response to point 10 in my previous review (R1), the authors seem to agree with my interpretation about integration and mention that the study is indeed about the integration of face and language cues, however, this isn’t clear from their description in the introduction or anywhere else in the manuscript. In fact, the word integration is not at all mentioned in the text. I would encourage them to elaborate further on this point – I think it will help them provide additional theoretical points of discussion later on in the text.

5. Now that I’ve seen the example trial provided in the Supplementary Materials, I believe that the authors will need to acknowledge the potential carryover effects. What I mean by carryover effects here, is the possibility that participants use the ratings they have provided for one dimension (say valence) to inform their rating on another dimension (say arousal), whereas ideally both ratings should be solely based on the label/face information only.

6. I understand what the authors’ are trying to say in their response to point 21. I don’t think I have misread the data in the appendix but rather misinterpreted the authors’ explanation. They argue that participants were not influenced by the labels for those emotions but looking at the appendix it seems that when seeing the labels alone participants were clearly influenced by the labels. Therefore, it is not that participants were not influenced by the labels but rather that the labels might not have been taken into consideration when paired with a face image. If this is what the authors meant to say here, I'd encourage them to make it clearer.

7. I’m still quite wary of the lack of differences between the face only and face+label conditions. I take the authors’ point about the difference between a within- and a between-subjects design but there were quite a few participants in both groups. I suspect the effect is weaker and that is why it cannot be detected with a between-subjects design. I think the authors need to be more upfront about this finding – I can see that they have mentioned it in the discussion but that is not enough to draw the reader’s attention to this important point. I’d suggest either elaborating on this point in the discussion (rather than just briefly mentioning it as an idea for future research) or reporting this in the results section (the full analysis can still remain in the Supplementary materials but the main outcome should be mentioned in the main text).

8. Throughout the analysis sections, the authors mention that the findings are consistent with the valence ratings for Labels Alone. It is not clear what this really means, especially when there are no statistical comparisons between these two conditions provided. Therefore, this seems rather speculative.

9. I would challenge the authors’ conclusions following Study 2 that there is a striking similarity between the results from Study 1 and 2 – they might have to clarify that this is for arousal and dominance only.

10. Supplementary materials will need to be organised in a more efficient way – I’d suggest assigning each separate section an appendix number/letter and referring to that section in the main text. It’s not clear why appendix A is not included in the supplementary materials.

11. My most major outstanding issue concerns the discussion. At the moment, it mostly reads like a summary of the results with a few very brief theoretical points being made. This was something mentioned by R3 as well and I think that’s still true for the current version of the manuscript. I have already suggested some interesting theoretical points of discussion both in my first review and here (moreover, some of the results suggest that labels might have a stronger effect on dynamic faces (especially for valence) which contradicts some of the existing literature. This might also be a potential point for discussion). I understand the authors’ point that speculating post hoc might be a poor research practice but simply summarising their results without much reference to the broader theoretical issues and the implications of their findings does not demonstrate the value of their work.

12. I also agree with Reviewer 3 that an additional study that allows for a more direct comparison between faces only, labels only and faces+labels would immensely strengthen the manuscript and its overall impact. It is a shame that the authors are unable to collect these data.

Reviewer #2: The authors have provided a thorough revision and addressed all of my comments. Congrats on the great paper!

Reviewer #3: Dear Authors,

Thank you for your thorough revision. I am impressed by the time and detail you made to improving your manuscript and incorporating the provided suggestions. There were only a few comments which were not executed (and would have required collecting more data), but you explained why this was not possible. Overall, I am satisfied with the changes you have made and recommend this manuscript for publication. I am very fond of this work, and it nicely aligns with my own.

Kind Regards,

7. PLOS authors have the option to publish the peer review history of their article (what does this mean?). If published, this will include your full peer review and any attached files.

Reviewer #1: No

Reviewer #2: Yes: Erik Nook

Reviewer #3: Yes: Jennifer M B Fugate

---

## [Author Response · Author response to Decision Letter 1]

11 Jun 2020

Re: PONE-D-19-26857R1

‘Grumpy’ or ‘Furious’? Arousal of Emotion Labels Influences Judgments of Facial Expressions (Barker, Bidstrup, Robinson and Nelson)

Thank you for the comments and for inviting us to submit a second revision to PLOS ONE. Two of the three reviewers were entirely satisfied with the first revision, but one reviewer had remaining concerns. For convenience, we reproduce Reviewer 1’s comments verbatim with our responses positioned under each comment. Changes made to the manuscript are in italic text.

Reviewer #1: 

1. This is not a concern, but I would like to mention that the introduction reads much better now and it motivates the two studies nicely.

• Thank you for the positive feedback.

2. I appreciate that the authors have provided a hypothesis regarding their predicted U-shaped curve describing the relationship between arousal, valence and dominance. It might be helpful to clarify exactly how they expect this to be demonstrated in the data from the present two studies. Looking at Figure 1, it seems like some of the results might not fit such a relationship – this should be discussed later on in the text.

• Upon reflection, the U-shaped relationship was not the best way to word our hypothesis. Therefore, we have clarified our hypothesis regarding valence and arousal, and added an example for the reader (p. 8): “We hypothesised that label arousal level would also influence the valence and dominance ratings of faces consistent with the valence and dominance of the label. For example, we expected that if a high arousal label was low in valence, the valence judgement of the paired facial expression would be lowered.”

3. I’m not entirely sure the power analysis provided by the authors is correct. A post hoc analysis provides information about the achieved power, not about the sample size needed to achieve that power. Moreover, since they are using a within-subjects design (2 conditions x 8 emotions), there should be only one group of participants.

• We agree with the reviewer that post-hoc power analyses are controversial (e.g., http://daniellakens.blogspot.com/2014/12/observed-power-and-what-to-do-if-your.html#:~:text=Observed%20power%20(or%20post%2Dhoc,true%20difference%20to%20be%20found.) and have removed the power analysis from the Supplemental Materials. However, our original justification of sample size (that is, group sizes that are similar to those used in previous research) remains in the manuscript.

4. In their response to point 10 in my previous review (R1), the authors seem to agree with my interpretation about integration and mention that the study is indeed about the integration of face and language cues, however, this isn’t clear from their description in the introduction or anywhere else in the manuscript. In fact, the word integration is not at all mentioned in the text. I would encourage them to elaborate further on this point – I think it will help them provide additional theoretical points of discussion later on in the text.

• As requested by this reviewer, we now make this point about “integration” clearer in the Introduction (p. 6) and the Discussion (p. 24). For example: “The current studies aim to investigate whether emotion word labels are integrated into judgments of emotion in facial expressions.”

5. Now that I’ve seen the example trial provided in the Supplementary Materials, I believe that the authors will need to acknowledge the potential carryover effects. What I mean by carryover effects here, is the possibility that participants use the ratings they have provided for one dimension (say valence) to inform their rating on another dimension (say arousal), whereas ideally both ratings should be solely based on the label/face information only.

• Although the pattern of results we found do not suggest carryover effects are driving our results, we now acknowledge this possibility in the discussion on pg. 25, stating “It is also possible that by rating the three dimensions at the same time, participants’ responses were influenced by their ratings of the other dimensions – future research might consider having participants rate each dimension in separate blocks”. 

6. I understand what the authors’ are trying to say in their response to point 21. I don’t think I have misread the data in the appendix but rather misinterpreted the authors’ explanation. They argue that participants were not influenced by the labels for those emotions but looking at the appendix it seems that when seeing the labels alone participants were clearly influenced by the labels. Therefore, it is not that participants were not influenced by the labels but rather that the labels might not have been taken into consideration when paired with a face image. If this is what the authors meant to say here, I'd encourage them to make it clearer.

• As suggested, the wording of this sentence has been changed to read: “label information did not appear to be integrated into ratings of happy and embarrassed expressions”. (p. 24) 

7. I’m still quite wary of the lack of differences between the face only and face+label conditions. I take the authors’ point about the difference between a within- and a between-subjects design but there were quite a few participants in both groups. I suspect the effect is weaker and that is why it cannot be detected with a between-subjects design. I think the authors need to be more upfront about this finding – I can see that they have mentioned it in the discussion but that is not enough to draw the reader’s attention to this important point. I’d suggest either elaborating on this point in the discussion (rather than just briefly mentioning it as an idea for future research) or reporting this in the results section (the full analysis can still remain in the Supplementary materials but the main outcome should be mentioned in the main text).

• As requested, these findings are now reported in the Results sections of the main text (p. 14, p. 18). 

8. Throughout the analysis sections, the authors mention that the findings are consistent with the valence ratings for Labels Alone. It is not clear what this really means, especially when there are no statistical comparisons between these two conditions provided. Therefore, this seems rather speculative.

• To clarify, we meant that the direction of the effect was consistent with what would be expected based on the labels of ratings alone. For example, surprised face + high arousal “astounded” was rated as more negative, while proud face with high arousal “victorious” was rated as more positive. This consistent with the valence ratings for Labels Alone. For clarity, we have changed the wording to read: “These findings are in the same direction as the valence ratings for Labels Alone” (e.g., p. 19, 20, 22)

9. I would challenge the authors’ conclusions following Study 2 that there is a striking similarity between the results from Study 1 and 2 – they might have to clarify that this is for arousal and dominance only.

• We have opted to remove this sentence from the manuscript (p. 20).

10. Supplementary materials will need to be organised in a more efficient way – I’d suggest assigning each separate section an appendix number/letter and referring to that section in the main text. It’s not clear why appendix A is not included in the supplementary materials.

• As requested, we have reorganised the supplementary materials and assigned each section an Appendix number. We have also moved the table previously in Appendix A to the Supplemental Materials.

11. My most major outstanding issue concerns the discussion. At the moment, it mostly reads like a summary of the results with a few very brief theoretical points being made. This was something mentioned by R3 as well and I think that’s still true for the current version of the manuscript. I have already suggested some interesting theoretical points of discussion both in my first review and here (moreover, some of the results suggest that labels might have a stronger effect on dynamic faces (especially for valence) which contradicts some of the existing literature. This might also be a potential point for discussion). I understand the authors’ point that speculating post hoc might be a poor research practice but simply summarising their results without much reference to the broader theoretical issues and the implications of their findings does not demonstrate the value of their work.

• We have broadened our discussion to highlight some of the theoretical points suggested by the reviewer – particularly the influence of dynamic vs. static expressions. We hope this additional discussion better highlights our findings, without too much post hoc speculation. 

12. I also agree with Reviewer 3 that an additional study that allows for a more direct comparison between faces only, labels only and faces+labels would immensely strengthen the manuscript and its overall impact. It is a shame that the authors are unable to collect these data.

• We agree with this reviewer that it is unfortunate we are unable to collect additional data.

Sincerely,

Megan Barker, Ph.D. (on behalf of co-authors)

---

## [Editor Report · Decision Letter 2]

16 Jun 2020

"Grumpy" or "Furious"? Arousal of Emotion Labels Influences Judgments of Facial Expressions

PONE-D-19-26857R2

Dear Dr. Barker,

We’re pleased to inform you that your manuscript has been judged scientifically suitable for publication and will be formally accepted for publication once it meets all outstanding technical requirements.

Kind regards,

Peter A. Bos

Academic Editor

PLOS ONE
---

## [Editor Report · Acceptance letter]

18 Jun 2020

PONE-D-19-26857R2 

"Grumpy" or "Furious"? Arousal of Emotion Labels Influences Judgments of Facial Expressions 

Dear Dr. Barker:

I'm pleased to inform you that your manuscript has been deemed suitable for publication in PLOS ONE. Congratulations! Your manuscript is now with our production department. 

Kind regards, 

on behalf of

Dr. Peter A. Bos 

Academic Editor

PLOS ONE